



# A Novel Approach to Estimate Carbon and Nitrogen Flux from In Situ Optics: Application to Cyclonic Eddies off the Cape Verde Islands

Nasrollah Moradi[1,2], Lili Hufnagel[1,2,*], Simon Ramondenc[1,2,§], Clara M. Flintrop[1,†], Rainer Kiko[3,4,5], Tim Fischer[6], Helena Hauss[7,8], Arne Körtzinger[9,5], Gerhard Fischer[2] and Morten H. Iversen[1,2]

[1]Alfred Wegener Institute Helmholtz-Center for Polar and Marine Research Bremerhaven, Bremerhaven, Germany
[2] MARUM and University of Bremen, Bremen, Germany
[3]Laboratoire d'Océanographie de Villefranche, LOV Institut de la Mer de Villefranche, IMEV, Villefranche-sur-Mer, France
[4]Department of Biological Oceanography, GEOMAR Helmholtz Centre for Ocean Research Kiel, Kiel, Germany
[5]Faculty of Mathematics and Natural Sciences, Kiel University, Kiel, Germany.
[6]Department of Physical Oceanography, GEOMAR Helmholtz Centre for Ocean Research Kiel, Kiel, Germany
[7]Division of Climate & Environment, Norwegian Research Centre (NORCE), Bergen, Norway
[8]Department of Ocean Ecosystems Biology, GEOMAR Helmholtz Centre for Ocean Research Kiel, Kiel, Germany
[9]Department of Chemical Oceanography, GEOMAR Helmholtz Centre for Ocean Research Kiel, Kiel, Germany
[*]Current address: Helmholtz Centre for Environmental Research - UFZ, Department of River Ecology, Magdeburg, Germany
[§]Current address: Sorbonne University, CNRS, Laboratoire d'Océanographie de Villefranche, Villefranche-sur-Mer, France
[†]Current address: The Fredy & Nadine Herrmann Institute of Earth Sciences, Hebrew University of Jerusalem, Jerusalem, Israel Interuniversity Institute for Marine Sciences, Eilat, Israel

*Correspondence to*: Nasrollah Moradi (nasrollah.moradi@awi.de) and Morten Iversen (morten.iversen@awi.de)

**Abstract.** Mesoscale eddies are pervasive in the global ocean and are suggested to play a significant role in enhancing carbon export from the surface to the deep ocean. However, their dynamic nature and associated challenges of in-situ sampling have limited direct measurements of carbon flux within these features, leaving their contribution to carbon export uncertain. In-situ optical devices offer a promising solution by providing high-resolution data on particle abundance and size distribution (PSD) within eddies, both vertically and horizontally. Yet, converting PSD data into reliable carbon flux 30 estimates is particularly challenging in dynamic features like eddies, and is further complicated by the need to account for key factors regulating carbon export. To address this, we present a novel mechanistic framework that optimally integrates PSD data with flux measurements, settling velocities and respiration rates of in-situ collected aggregates, together with water temperature and oxygen concentration to estimate both particulate carbon and nitrogen fluxes in the water column. This framework tackles primary challenges by incorporating contributions from suspended particles, size-dependent sinking 35 velocities, and the degradation rates of settling particles with depth. Applied to a multi-instrument dataset from a high-resolution survey of eddies off the Cape Verde Islands, the presented framework reveals distinct funnel-shaped flux patterns for particulate organic carbon and nitrogen, with a twofold enhancement of flux within the examined cyclonic eddy. Furthermore, it identifies lower carbon-to-nitrogen ratios in settling organic matter at the deep eddy core, indicating the transport of fresher organic matter to the deep sea. These findings provide new insights into eddy-driven carbon export, 40 highlighting the roles of settling velocity, water temperature, and oxygen concentration in shaping carbon flux and organic



matter stoichiometry. While observed in a single eddy, this doubling of carbon flux underscores the potential for mesoscale eddies to locally enhance carbon export and, given their widespread occurrence, contribute significantly to the ocean's capacity to sequester atmospheric CO2.

## 1 Introduction


The eastern tropical North Atlantic (ETNA) is known for its high-productivity and low-oxygen eddies, which typically are anticyclonic mode-water and cyclonic eddies (Karstensen et al., 2015, 2017; Fiedler et al., 2016; Schütte et al., 2016a, 2016b; Pietri and Karstensen, 2018; Dilmahamod et al., 2022). When these eddies form in coastal productive waters and

migrate to oligotrophic regions, they become oases of elevated biological activities and may result in high trophic transfer (Christiansen et al., 2018) and carbon export to the deep ocean (Fiedler et al., 2016, Romero et al., 2016; Hauss et al. 2015; Fischer et al., 2016, 2021). However, the spatial variability of carbon export within eddies necessitates high-resolution flux measurements of organic matter, both horizontally and vertically, to better understand their contribution to carbon export to the deep ocean. While sediment traps provide direct measurements of organic matter flux, they collect bulk samples of

sinking particles from a limited area and depth (Buesseler et al., 2007, 2008). Furthermore, flux measurements using drifting sediment traps (DST) or neutrally buoyant sediment traps in deeper layers of eddies are rare. In contrast, in-situ camera systems (hereafter ISC) are operationally simple and can obtain high-resolution spatiotemporal observations of particle abundance and size distribution (PSD), including settling aggregates and zooplankton (Guidi et al., 2008; Iversen et al., 2010; Picheral et al., 2010; Kiko et al., 2022). ISC-derived depth-specific particle concentrations can serve as proxies for

local particle flux in the water column. As such, calibration of the ISC-derived PSD from depths with corresponding DST flux allows for estimates of size-specific carbon flux associated with aggregates, which can be upscaled to have high vertical resolution of particulate organic matter (POM) flux through the water column (Guidi et al., 2008; Iversen et al., 2010; Ramondenc et al., 2016). In a well-established statistical approach proposed by Guidi et al. (2008) for this calibration, hereafter referred to as the conventional method, particulate organic matter (POM) fluxes in the water column are

parameterize as $\sum n_i A d_i^B$. Here, $n$ denotes the ISC-derived depth-specific particle concentration as a function of particle size ($d$), with the summation performed over all considered particle size bins, indexed by $i$. The free parameters $A$ and $B$ are empirically optimized through related DST-based flux measurements.

A common challenge when using ISC-derived particle profiles to estimate biogeochemical fluxes is distinguishing between settling and suspended particles in ISC images (McDonnell and Buesseler, 2012; Cael and White, 2020), as including

suspended particles leads to an inaccurate estimate of sinking POM flux. Sinking velocity and carbon degradation rates of settling aggregates are crucial factors that, along with zooplankton grazing, influence the strength and efficiency of the biological carbon pump and, consequently, the flux of POM through the water column (Asper et al., 1992; Simon et al., 2002; Iversen et al., 2010, 2013, 2015; Stukel et al., 2018; Moradi et al., 2018, 2021; Iversen, 2023). However, besides the contribution of suspended particles, these two factors are not explicitly considered in the conventional method, either as



input parameters to be measured regionally or as free parameters to be adjusted. Furthermore, in regions lacking DST-based flux measurements from either the upper or lower parts of the water column, the conventional method may not accurately optimize the parameters *A* and *B*. This can lead to overestimation of carbon flux at depth if only shallow DSTs are available, and vice versa. In this study, we address all these issues to more accurately determine particulate organic carbon (POC) and nitrogen (PON) fluxes through the entire water column. We achieve this by developing a novel approach that effectively

integrates in-situ camera particle profiles, associated DST-based flux observations, measured particle settling velocities and respiration rates of in-situ collected aggregates, together with CTD measurements of water temperature and oxygen concentration (Fig. 1). Our approach is particularly useful for quantifying POC and PON flux fields across oceanic dynamic features, such as mesoscale eddies, where flux measurements from deep waters are often lacking. As a case study, we apply this approach to a multi-instrument dataset from a high-resolution survey of eddies (Meteor 160), highlighting differences in

POM flux fields associated with cyclonic eddies and non-eddy features in the Cape Verde region (Fig. 2).

The remainder of the paper is structured as follows: Sect. 2 describes in-situ observations and laboratory measurements, along with the two methods developed to optimally integrate the observational data. The results are presented in Sect. 3, discussed in Sect. 4, and concluded in Sect. 5.

## 2 Materials and Methods

### 2.1 Shipboard-based measurements

*Drifting Sediment Traps (DST).* A total of six DST deployments were done, four (DST1, DST4-6) were southwest and two (DST2 and DST3) northeast of the Cape Verde Islands (see Fig. 2 for the position of examined stations). The DST array consisted of a top buoy equipped with an Iridium satellite sender, an array of 14 two-liter buoyancy spheres that acted as wave attenuators, two benthos spheres (glass floats) that provided 25 kg of buoyancy each, and three sediment trap arrays

each equipped with four gyroscopically mounted trap cylinders (of diameter 10 cm) deployed at 100 m, 200 m, and 400 m water depth, respectively. One trap cylinder per collection depth was fitted with a gel-filled collection cup with an ethanol-based viscous cryo-gel (Tissue-Tek, O.C.T.TM COMPOUND, Sakura) to collect and preserve settling aggregates, allowing determination of the physical structure, type, and size of individual aggregates. The three remaining trap cylinders at each collection depth were used to quantify biogeochemical fluxes (Pauli et al. 2021), including particulate organic carbon (POC)

and particulate organic nitrogen (PON).

*Underwater Vision Profiler (UVP).* During the expedition a UVP5 camera system (Picheral et al., 2010) was mounted on the CTD-Rosette system and deployed at 71 stations to quantify the vertical abundance and size distribution of particles and aggregates (Fig. 2). This included parallel deployments with each deployment and recovery of DST, allowing for

simultaneous observations by both the DST and UVP5. The UVP5 provides data on particles larger than 64 µm in equivalent spherical diameter (ESD). However, due to its pixel size of ∼30 µm, counts of particles from 120 µm upwards are generally



considered to be reliable. The majority of the profiles were done down to 1000 m depth and the size-specific particle concentrations were calculated for 5 m depth bins. The UVP5 particle data and the corresponding CTD measurements of water temperature and oxygen concentration have been published in the PANGAEA data repository by Kiko et al. (2022) and Dengler et al. (2022), respectively.

*Marine Snow Catcher (MSC).* Eight MSC (OSIL, UK) deployments were conducted to collect in-situ formed marine aggregates, which were used to measure size-specific sinking velocity of settling particles. The in-situ formed aggregates were collected from 10 m below the chlorophyll maximum. After deployment, aggregates were allowed to sink in the MSC for three hours before they were gently collected from the base of the MSC. A total of 95 individual particles were collected, and their size-specific sinking velocities were measured on board using a flow chamber system equipped with an oxygen microsensor (Ploug and Jørgensen, 1999). The flow chamber was filled with the filtered sea water sampled by the MSC. A heating system was used to adjust the water temperature of the flow chamber to match the in-situ temperature, as indicated by the CTD measurement of temperature profile from the collection depth for each station. The size-specific sinking velocities of 88 (out of 95) aggregates were fitted by a power law function of the form $W = A_w d^{B_w}$ where $A_w$ and $B_w$ are the fitting parameters, $d$ is the equivalent spherical diameter (ESD, mm) of particles, and $W$ is the settling velocity (m d$^{-1}$). Seven particles were excluded: one particle (6.1 mm) was larger than the upper size-range considered in this study (64 µm - 4.1mm) for calculating POC flux, and the remaining six were excluded due to a lack of replicate measurements of sinking velocity. During the measurements of the sinking velocities of the collected aggregates in the flow chamber, the oxygen concentration profile across the water-aggregate interface of 31 randomly selected aggregates was measured using oxygen microsensor, following the procedure described in Iversen and Ploug (2013). The microsensor used was a Clark-type oxygen microelectrode with a guard cathode (Revsbech, 1989) and a tip diameter of 8-12 µm, and was calibrated in oxygen-saturated and oxygen-depleted water before mounting on a micromanipulator. The concentration was measured in increments of 50 µm with a 3 sec waiting period before and a 3 sec measuring period during each measurement. The current was measured on a multimeter (Unisense) with a 90% response time of <35 msec. The interfacial oxygen flux for each aggregate was then calculated following Moradi et al. (2021). The average water temperature in the flow chamber during microsensor measurements across different stations was $18.10 \pm 0.18$ °C. An example of a measured $O_2$ concentration profile and its calculated corresponding diffusive flux is shown in Fig. A1.

*Gel trap image analysis.* Aggregates collected in the gel cups were imaged using a high-resolution camera (2.8 µm px$^{-1}$) to determine the size distribution of particle flux at depths 100, 200, and 400 m for each station. To ensure clear imaging of both small and large particles, we imaged smaller subsections of each cup area at three prespecified depths of camera's focus. This process was automated using a pre-programmed XYZ-stage equipped with the camera and a motor. A custom Python package, Gel-Trap Particle Image Size Analyzer (Gel-PISA), was developed to reconstruct a single high-resolution image of the entire gel cup area. This tool facilitates precise stacking and stitching of subsection images from each gel cup



(Fig. 3) and is specifically tailored for analyzing images captured by our camera system. Gel-PISA also removes background noise from each image frame, producing clean, high-quality images of individual particles. The removal of background image was achieved by analysing the grayscale value of each pixel at the three focus depths. For that, we leveraged the fact that the variation in grayscale values of pixels in the uncovered areas on the image (devoid of particles) across the three

focus depths is different from those in the covered areas. Subsequently, the final gel trap image underwent conversion into a binary black and white format. Particle size was calculated from its image area, determined using the "contour" module of "opencv-python" software. This allowed for accurate quantification of particle sizes and counts within the gel trap samples.

*Gel trap-based particle fluxes.* The equivalent spherical diameter (ESD) of all particles that were collected in the gel traps

was calculated using the formula $2\sqrt{\mathbb{A}/\pi}$, where $\mathbb{A}$ represents the pixel area of the particle multiplied by the pixel size of the image. The swimmers, i.e., zooplankton that actively entered the trap, were visually annotated and excluded from the calculations. Given the rarity of larger particles, particularly at greater water depths, it was necessary to group them into larger size bins. The particles were sorted into logarithmically spaced size bins such that the equivalent spherical volume corresponding to the particle sizes at the left and right edges of each bin increases by 50% from one bin to the next (Jackson

et al., 2005). Specifically, the bins were defined so that the right edge of each bin ($D_{i+1}$) is $\sqrt[3]{2}$ times larger than its left edge ($D_i$), i.e., $D_{i+1} = \sqrt[3]{2}D_i$ with $D_1 = 32\ \mu m$. The index $i$ refers to the $i^{th}$ size bin represented by $d_i = 0.5 \times (D_{i+1} + D_i)$, the midpoint of the size bin $i$. The size bin $i$ contains the number of particles whose diameters are within the range $D_i \leq d < D_{i+1}$. Using the deployment time of the sediment trap and the area of the gel trap cup (ca. 50 cm$^2$), the corresponding gel trap-based size distribution of particle flux ($n_{gel}^F$) was calculated in units of number of particles per square meter per day

(# m$^{-2}$d$^{-1}$). Each $n_{gel}^F$ was characterized by the best linear fit of $\log_{10}(n_{gel}^F)$ versus $\log_{10}(d)$, represented by an associated pair of intercept ($a^{gel}$) and slope ($b^{gel}$), as exemplified in Fig. 4a. For the fitting procedure, a particle size-range from 64 µm to 2.58 mm was considered. The midpoints of the smallest (64–80 µm) and largest (2.05–2.58 µm) size bins within this range are 72 µm and 2.315 mm, respectively. Particle counts for sizes larger than 2.58 mm were considered unreliable due to the rarity of large sinking particles, especially at greater depths, and the small sampling area by the trap. If a size bin in the

considered particle size range had $\log_{10}(n_{gel}^F) \leq 0$ (or simply no particles in that size bin), that size bin was excluded from fitting procedure. Using the characteristic $a^{gel}$ and $b^{gel}$ for each gel trap, the estimate of $n_{gel}^F$ as a function of the size bin is given by $n_{gel}^F(d_i) \approx 10^{a^{gel}} \times d_i^{b^{gel}}$.

*UVP-based particle concentration.* To determine UVP-based size distribution of particle concertation ($n_{uvp}^C$) the same size

bins used for the gel traps were applied, but the first three bins were skipped and $D_1$ was set to 64 µm because of lower resolution of UVP camera. Due to the UVP's shorter observation time and the notable decrease in particle counts with increasing water depth, particularly for larger particles at greater water depth, a depth-specific averaging approach was



implemented to statistically enhance the reliability of aggregate abundance and the size distribution. Averaging was conducted within specific vertical water layers: 5-meter layers for depths shallower than 50 m, 10-meter layers for depths

between 50 m and 200 m, 15-meter layers for depths between 200 m and 400 m, and 25-meter layers for greater depths. To associate a spatiotemporally consistent $n_{\mathrm{uvp}}^{\mathrm{C}}$ counterpart to each $n_{\mathrm{gel}}^{\mathrm{F}}$, the average of two $n_{\mathrm{uvp}}^{\mathrm{C}}$ were used. These two were extracted from the UVP profiles obtained at the times of deployment and recovery of the corresponding DST, aligning with the associated depth of respective $n_{\mathrm{gel}}^{\mathrm{F}}$.

*Eddy location and 3D-structure.* To locate survey stations and ship tracks relative to eddy coordinates, and to characterize the eddy dynamical regime, we used in-situ observed currents from the ship's vessel-mounted Acoustic Doppler Current Profiler (vmADCP). During Meteor cruise M160, an RDI Ocean Surveyor 75 kHz and an RDI Ocean Surveyor 38kHz (the latter only for the first half of the cruise) continuously recorded currents down to 600 m and 1200 m depth at 8 m and 32 m vertical bin size, respectively. The broadband mode that we used resulted in 1 minute-average velocity data of 0.03 m s$^{-1}$

precision at a high ping frequency (about 30 min$^{-1}$) under calm sea conditions (Dengler et al., 2022). These velocity records were used to perform an optimum fit to a simplified eddy model. Here, for each depth layer of the velocity time series (in our case 8 m depth strata), we assumed that the presence of an eddy during the period of observation should manifest in a circular symmetric flow field around a center and should slowly propagate as a whole at a constant drift. The quality function to be minimized is the remaining variance of the velocity time series after removal of the circular symmetric

propagating feature. Thus, after optimum search, we obtained the eddy center position and the drift velocity for each depth layer during observation time. Based on this 3 D-localization, a complete eddy flow field can be estimated, as a function of distance from the center points. Finally, dynamical variables like divergence and rotation (vorticity) can be derived from the azimuthal (tangential, swirl) and radial velocity components. The location of the eddy rim at each depth level could then be calculated as the radius of vorticity equal to zero, as this property can be used to estimate the trapping limit of the eddy

(Early et al., 2011). The eddy localization and structure reconstruction were well applicable for the eddy northeast of Sal Island (Fig. 2, zone A2), referred to as the Sal eddy, even though this eddy was rather weak. It was located at about 20.4°W 17.8°N on December 1, 2019, drifting towards southwest at 0.03 m s$^{-1}$ (2.6 km d$^{-1}$), maximum azimuthal velocity 0.08 to 0.11 m s$^{-1}$ (7.0 - 9.5 km d$^{-1}$), its rim at about 50 km radius, convergent flow field in 50 m to 150 m depth in inner 20 km to 25 km radius, causing upwelling and divergence above. Unfortunately, the eddy southwest of Brava Island (Fig. 2, zone A3),

which had been strong and near circular until immediately before sampling, showed unsteady behavior, sudden warming of the surface, and seemed quite excentric/elliptic just after the sampling started. So, the vital assumptions (circular symmetry and steadiness) seemed too violated to perform localization for this time period. Still, we can report the following characteristics based on the observed velocity field and deployed surface drifters (Carrasco et al., 2020): elliptical shape with long axis in NNE-SSW direction, long half axis about 50 km, short half axis about 20 km, 'center' at about 25.5°W 14.5°N,

maximum azimuthal velocity about 0.4 m s$^{-1}$. However, we cannot report what happened after our visit, whether re-





stabilization or collapse. Given that this eddy-like feature did not exhibit typical eddy characteristics at the time of sampling, we refer to it as the Brava eddy-like event.

**2.2 Data integration methods**

**2.2.1 Method I: Using particle size spectra from UVP and gel traps to exclude non-settling particles**

The obtained $n_{\text{gel}}^{\text{F}}$ and $n_{\text{uvp}}^{\text{C}}$ have different physical dimensions; $n_{\text{gel}}^{\text{F}}$ represents particle flux [#/(area×time)], while $n_{\text{uvp}}^{\text{C}}$ represents particle concentration [#/volume]. To enable comparison, $n_{\text{uvp}}^{\text{C}}$ for each size bin was multiplied by the associated sinking velocity ($W$) to convert it into a UVP-based SDPF ($n_{\text{uvp}}^{\text{F}}$):

$$n_{\text{uvp}}^{\text{F}}(d_i)[\#/\text{m}^2\text{day}] = n_{\text{uvp}}^{\text{C}}(d_i)[\#/\text{m}^3] \times W(d_i)[\text{m/day}] \tag{1}$$


where $d_i$ denotes the midpoint of the $i^{\text{th}}$ size bin. Each particle flux size distribution was characterized by an intercept and slope, obtained through a best linear fit of $\log_{10}(n^F)$ versus $\log_{10}(d)$ (see Fig. 4b for an example). The logarithmic transformation incorporates contributions from both small and large particles. However, the calculated $n_{\text{uvp}}^{\text{F}}$ tended to overestimate actual particle fluxes because the UVP camera, unlike sediment traps, captures both suspended and settling

particles (see Fig. 4c for an illustration). Adjustments to their characteristic intercept-slope pairs were therefore necessary to use them as a reliable proxy for biogeochemical flux estimates in the water column. We assumed that $n_{\text{gel}}^{\text{F}}$ provided reliable flux estimates of the size distribution of settling particles, since suspended particles do not enter sediment traps and the swimmers are manually removed during image analysis. The characteristic gel trap-based slope–intercept pairs ($a^{\text{gel}}$ and $b^{\text{gel}}$) served as references for optimizing the correction factors α and β, which adjust the associate intercepts and slopes of

$n_{\text{uvp}}^{\text{F}}$ as $\alpha \times a^{\text{uvp}}$ and $\beta \times b^{\text{uvp}}$, respectively, through a geometrical optimization approach. A trapezoid was constructed for each SDPF based on the characteristic line of best fit and the considered particle size range. The area of this trapezoid (Fig. 4d), denoted as S, was calculated separately for each $n_{\text{gel}}^{\text{F}}$ and for its UVP-based counterpart, $n_{\text{uvp}}^{\text{F}}$. The correction factors were optimized by minimizing the sum of squared error term ($S_{\text{error}}$), defined as $S_{\text{error}} = \sum(S_k^{\text{gel}} - S_k^{\text{uvp}})^2$, to best align the associated trapezoidal areas of $n_{\text{uvp}}^{\text{F}}$ with those of $n_{\text{gel}}^{\text{F}}$. Here, $S_k^{\text{gel}}$ and $S_k^{\text{uvp}}$ represent the geometric areas of the $k$-th gel trap-

based particle flux and its UVP counterpart, respectively, and the summation is over all the gel trap samples. The areas of the resulting trapezoids were calculated as follows:

$$S^{\text{gel}} = L \times (y_1^{\text{gel}} + y_2^{\text{gel}})/2 \text{ where } L = x_2 - x_1, \ y_1^{\text{gel}} = a^{\text{gel}} + b^{\text{gel}}x_1, \text{ and } y_2^{\text{gel}} = a^{\text{gel}} + b^{\text{gel}}x_2. \tag{2}$$

Similarly,





$$S^{uvp} = L \times (y_1^{uvp} + y_2^{uvp})/2 \text{ where } y_1^{uvp} = \alpha a^{uvp} + \beta b^{uvp} x_1 \text{ and } y_2^{uvp} = \alpha a^{uvp} + \beta b^{uvp} x_2. \tag{3}$$

In both cases, $x_1$ and $x_2$ are fixed to $\log_{10}(144)$ and $\log_{10}(2315)$ respectively, representing overlapping size range where SDPF from both the UVP5 and gel traps provide reliable slopes. These sizes correspond to the midpoints of the 128–161 µm and 2050–2580 µm size bins, respectively. Note that the particle fluxes ($n_{gel}^F$ and $n_{uvp}^F$) and particle size $d$ were initially normalized by 1 µm and 1 m$^{-2}$d$^{-1}$, respectively, before logarithmic transformation to conveniently perform calculations using dimensionless numbers. Therefore, the calculated intercepts, slopes, and trapezoidal areas are unitless.

The Python "scipy" library's "fmin" function was applied to minimize $S_{error}$, determining optimal values for $\alpha$ and $\beta$. To avoid local minima, "fmin" was run 1000 times with different initial values, ensuring convergence to the global minimum within the specified search space. This process yielded the best correction factors, $\alpha$ and $\beta$, aligning the intercepts and slopes of the fitted lines of $\log_{10}(n_{uvp}^F)$ to best match their gel trap-based counterparts. Note that these corrections were applied to the $\log_{10}(n_{uvp}^F)$ data. Therefore, to estimate the corrected UVP-based SDPF, we needed to convert them back to the normal scale as $n_{uvp}^{F*} = 10^{a^*} \times d^{b^*}$, where $a^* = \alpha \times a$ and $b^* = \beta \times b$, with $d$ in µm and $n_{uvp}^{F*}$ in # m$^{-2}$ d$^{-1}$. The $n_{uvp}^{F*}$ refers to the UVP-based SDPF after correction, excluding contributions from suspended particles and swimmers—collectively referred to as non-settling particles—detected by the UVP.

**2.2.2 Method II: Estimating POC and PON fluxes from UVP particle profiles and DST flux observations**

Each size bin was associated with a spherical "model particle" of diameter $d$ (mm), assigned a certain quantity of organic carbon mass $M^{car.}$ (mg) and a sinking velocity $W$ (m d$^{-1}$) derived from the size-velocity relationship of in-situ collected settling aggregates. Initial values for $M^{car.}$ were randomly selected and then optimized using DST-based flux measurements as described below. As these model particles sink, they undergo carbon degradation due to respiration by hosted bacterial communities. We assumed that the initial carbon content of the particle at the surface water is related to its size through a power function: $M^{car.}(\ell = 0) = A_{car.} d^{B_{car.}}$ where $A_{car.}$ and $B_{car.}$ are parameters to optimize, with surface water depth $\ell$ set to zero. However, as depth increases, the carbon content decreases due to microbial degradation. The carbon content at a given depth $\ell$ can thus be expressed as $M^{car.}(\ell) = A_{car.} d^{B_{car.}} - \lambda \times t$, where $\lambda$ is the carbon degradation rate (mg d$^{-1}$) and $t$ is the size-specific particle's travel time (in days) from the surface water to depth $\ell$, estimated as $\ell/W$. Given that degradation rate depends on temperature $T$ (°C) and dissolved oxygen concentration $O_{2,con.}$ (µmol $O_2$ kg$^{-1}$) (Iversen and Ploug 2013; Ploug, and Bergkvist 2015; DeVries and Weber 2017; Moradi et al., 2018)—both of which vary with water depth—we assumed a functional relationship for particle carbon content as:

$$M^{car.} = A_{car.} d^{B_{car.}} - \lambda \times t \quad \text{with} \quad \lambda = \lambda_{ref} \times Q_{10}^{(T-T_{ref.})/10} \times \left(\frac{O_{2,con.}}{K_m + O_{2,con.}}\right). \tag{4}$$





In this equation, $\lambda_{\text{ref}}$ is the carbon degradation rate at a reference temperature $T_{\text{ref}}$, $Q_{10}$ represents the rate at which degradation decreases with a 10°C temperature drop, and $K_m$ is the oxygen half-saturation constant. As direct measurements

of $Q_{10}$ and $K_m$ were unavailable for our study area, we adopted optimized values from DeVries and Weber (2017): $Q_{10}$=2.5 and $K_m$= 19 µmol $O_2$ kg$^{-1}$. Directly measuring $\lambda_{\text{ref}}$ for sinking aggregates is extremely challenging, so we approximate it using aggregate microbial respiration rates ($Q_{O_2}$) as $\lambda_{\text{ref}} = Q_{O_2} \times \emptyset \times \sigma$, where $Q_{O_2} = J_{O_2} \times \Lambda$ (25). Here, $J_{O_2}$ (µmol $O_2$ mm$^2$ d$^{-1}$) is the measured oxygen flux at the interface of in-situ collected particles, $\Lambda = \pi d^2$ (mm$^2$) represents particle surface area, $\emptyset$=1 is the stoichiometric ratio for oxygen-to-carbon conversion (assuming a respiratory quotient of 1 mol $O_2$ to

1 mol $CO_2$), $\sigma = 12$ is the molar mass of carbon in grams. With this formulation, $\lambda_0 \equiv J_{O_2} \times \emptyset \times \sigma$ (mg mm$^2$ d$^{-1}$) can be interpreted as the surface-area-normalized organic carbon degradation rate of particles, driven by the microbial respiration.

Bringing everything together, the estimation of POC flux based on $n_{\text{uvp},i}^{\text{F}^*}$ (calculated using Method I) at a given station and depth $\ell$ is expressed as:

$$\text{POC}^{\text{es.}} = \sum_i n_{\text{uvp},i}^{\text{F}^*} \times M_i^{\text{car.}} = \sum_i n_{\text{uvp},i}^{\text{F}^*} \times (A_{\text{car.}} d^{B_{\text{car.}}} - \lambda_{i,\text{ref.}} \times Q_{10}^{(T-T_{\text{ref.}})/10} \times (\frac{O_{2,\text{con.}}}{K_m + O_{2,\text{con.}}}) \times t_i) \qquad (5)$$

where the index $i$ refers to the $i^{\text{th}}$ size bin and the summation is over all the size bins considered. This approach for estimating POC flux incorporates a temporal component, which directly allows degradation of organic carbon with depth in the model particles and thus the associated POC flux attenuation across the entire water column. Depending on the particle

size, water temperature and oxygen concentration, the model particles might undergo complete carbon degradation at some specific depth. To accommodate this, the $M_i^{\text{car.}}$ was set to zero for all depths below the depth of complete degradation for the model particle of that size bin. The next step was to determine the optimization parameters by minimizing the sum of squared error term, following the approach of Guidi et al. 2008:

$$\text{Error} = \sum_k [\log_{10}(\text{POC}_k^{\text{m.}}) - \log_{10}(\text{POC}_k^{\text{es.}})]^2 \qquad (6)$$

where $\text{POC}_k^{\text{m.}}$ and $\text{POC}_k^{\text{es.}}$ are respectively the $k^{\text{th}}$ measured sediment trap-based POC flux and its corresponding UVP-based estimate, both expressed in units of mg m$^{-2}$ d$^{-1}$, and the summation is over all the measured POC fluxes (the $\text{POC}_k^{\text{es.}}$ values were calculated using Eq. 5).

To formulate the estimated PON flux (PON$^{\text{es.}}$), we assumed a correlation between the organic nitrogen content of the settling particles ($M^{\text{nit.}}$) and $M^{\text{car.}}$, as evident from the in-situ flux observations (see Sect. 3.2). The correlation is expressed as $M^{\text{nit.}} = \gamma \times M^{\text{car.}}$, where $\gamma$ is the unitless correlation factor and the resulting $M^{\text{nit.}}$ is in units of mg. Since nitrogen is a





limited element in the ocean and is typically degraded faster than carbon in the sinking particles, i.e., there is a preference for nitrogen over carbon utilization by aggregate-associated bacteria (Grossart and Ploug 2001), we treat $\gamma$ to be a function of

the particle travel time from the surface water to the respected water depth and consider a power function form for it as $\gamma = A_{\text{nit.}} t^{B_{\text{nit}}}$, where $t$ is the particle's travel time already defined, and $A_{\text{nit.}}$ and $B_{\text{nit.}}$ two parameters to be optimized. With this, the PON flux at a given station and depth $\ell$ is estimated as:

$$\text{PON}^{\text{es.}} = \sum_i n_{\text{uvp},i}^{\text{F}^*} \times M_i^{\text{nit.}} = \sum_i n_{\text{uvp},i}^{\text{F}^*} \times A_{\text{nit.}} \, t_i^{B_{\text{nit.}}} \times M_i^{\text{car.}} \tag{7}$$


In which the index $i$ refers the $i^{\text{th}}$ size bin and summation is over all the size bins considered. Similar to the estimation of POC flux, the parameters $A_{\text{nit.}}$ and $B_{\text{nit.}}$ were determined by minimizing the associated the sum of squared error term:

$$\text{Error} = \sum_k [\log_{10}(\text{PON}_k^{\text{m.}}) - \log_{10}(\text{PON}_k^{\text{es.}})]^2 \tag{8}$$


where $\text{PON}_k^{\text{m.}}$ and $\text{PON}_k^{\text{es.}}$, both in units of mg m$^{-2}$ d$^{-1}$, are respectively the $k^{\text{th}}$ measured sediment trap-based PON flux and its corresponding UVP-based estimate. The summation is over all the measured PON fluxes, with $\text{PON}_k^{\text{es.}}$ calculated using Eq. 7.

The dimensions of all parameters defined in the methodology are listed in Table A1.

## 3 Results

### 3.1 Size-velocity relationship and oxygen diffusive fluxes of in-situ collected aggregates

While the highly heterogeneous composition and different shapes of marine aggregates imply that the particle size alone cannot fully account for the variation in sinking velocities (Iversen and Lampitt, 2020), it remains the most practical parameter that can be readily measured in-situ on individual particles (Cael et al., 2021). Consequently, particle size is often

employed to predict the sinking velocity of settling marine aggregates. The size-specific sinking velocities of in-situ collected aggregates from the study area were parameterized as $W(d) = A_w d^{B_w}$, with the optimal parameters $A_w = 120.99$ and $B_w = 0.5005$ determined via least-squares regression (Fig. 5a). This implies that the average sinking velocity for particles sized 1 mm in this region is ca. 120 m d$^{-1}$. Using this relationship, the sinking velocities for the considered particle size bins are illustrated in Fig. 5b, where the midpoint of each bin predicts its associated sinking velocity. The smallest size bin (64–

80 µm) corresponds to a sinking velocity of 33 m d$^{-1}$, while the largest size bin (3.25–4.1 mm) to a sinking velocity of 231 m d$^{-1}$. These values delineate the typical velocity range of sinking particles for the specified size range in this region. Notably, the size-velocity relationship observed in the region aligns with results from a comprehensive size-velocity dataset (N=5655) compiled from 54 different studies (Fig. 5c), which presents an overall size-velocity relationship by increasing the



degree of variability (Cael et al., 2021). This suggests that the sinking aggregates collected off Cape Verde, although small in
sample size (N=88), represent a relatively heterogeneous pool of aggregates, a characteristic also evident in the gel trap
images (Fig. 3d). The microbial respiration-driven diffusive oxygen flux at the water–aggregate interface of the collected
aggregates is depicted in Fig. 5d. As seen, there is no evident correlation between the interfacial diffusive oxygen flux and
particle size. The mean of measured fluxes was $J_{O_2}$= 1.15 ± 0.16 nmol $O_2$ mm$^{-2}$ d$^{-1}$, resulting in a surface-area–normalized
particle carbon degradation rate of $\lambda_0 = 1.38 \times 10^{-5}$ mg C mm$^{-2}$ d$^{-1}$.


**3.2 DST-based POC, PON fluxes, and C:N ratios**

The measured fluxes of POC and PON, along with their corresponding molar C:N, ranged from 35.5 to 105.4 mg C m$^{-2}$ d$^{-1}$,
5.1 to 14.9 mg N m$^{-2}$ d$^{-1}$, and C:N from 5.7 to 11.4, respectively (see individual measurements in Table A2). Across all
stations, the highest POC and PON fluxes occurred at 100 m depth, with significantly higher flux than at 200 m (Wilcoxon
paired rank sum test, $p$ = 0.0064), meaning that in general there was a strong attenuation between these depths (Fig. 6a).
The strongest relative POC flux attenuation between 100 and 200 m was observed for DST3 (53.1%), which was deployed at
the rim of Sal eddy. At depths below 200 m, we did not observe a strong attenuation and the average POC fluxes at 200 m
and 400 m were not significantly different (Wilcoxon paired rank sum test, $p$ = 0.74877). A similar trend was observed for
the measured PON fluxes (Fig. 6b). Overall, the corresponding average C:N increased with increasing water depth (Fig. 6c).
A positive correlation was observed between individual POC and PON flux measurements (Fig. 6d).

**3.3 Size distribution of particle flux (SDPF) derived from gel traps and UVP using Method I**

The intercept and slope of the fitted lines characterizing the log-transformed SDPF (Fig. 7a, c; Table A2) were within the
ranges of $8.61 \leq a^{gel} \leq 11.13$ and $-2.69 \leq b^{gel} \leq -1.65$ for the gel traps, and $9.60 \leq a^{uvp} \leq 12.44$ and $-2.83 \leq b^{uvp} \leq -1.88$
for their UVP counterparts (calculated using Eq. 1). Overall, the $a^{uvp}$ (mean value: 10.55 ± 0.15) were higher than $a^{gel}$
(mean value: 9.77 ± 0.14), and $b^{uvp}$ (mean value: –2.25 ± 0.05) were steeper than $b^{gel}$ (mean value: –2.12 ± 0.06). To
account for these deviations, primarily due to the suspended particles detected by the UVP camera, the correction factors $\alpha$
and $\beta$ were optimized (see Sect. 2.2.1) as $\alpha$ = 0.9269 and $\beta$ = 0.9426 and applied to correct the UVP-based characteristic
intercepts and slopes as $a^* = \alpha \times a^{uvp}$ and $b^* = \beta \times b^{uvp}$ , respectively (Fig. 7b, d; Table A2).


**3.4 Estimating POC and PON flux from UVP-derived particle profiles using Method II**

Minimizing the error function in Eq. 6 and Eq. 8 for the particle size range considered in this study (64 µm–4.1 mm) yielded
optimized values of $A_{car.}$ = 1.1074 ×10$^{-3}$, $B_{car.}$ = 1.9286 for estimated POC (Eq. 5), and $A_{nit.}$= 6.5477 and $B_{nit.}$= 0.1677 for





estimated PON (Eq. 7) fluxes. The resulting average errors relative to the measured DST-based fluxes were 19.5% for POC

and 24.2% for PON (Fig. 8a–l). Figure 8m–r shows the estimated molar POC to PON ratios (C:N = $(14/12) \times$ POC$^{\text{es.}}$/ PON$^{\text{es.}}$)), compared to the corresponding DST-based measurements. With a relative average error of 17.35%, the estimated C:N not only align closely with the measured DST-based C:N but also display a general increase with depth, consistent with the profiles obtained from the DST-collected material (Fig. 6c). It is worth emphasizing that, in contrast to estimated POC and PON fluxes, we did not optimize any fitting parameters for the estimated C:N. However, we still found a better match

between estimated and measured C:N than we did for both POC and PON fluxes. For comparison, we applied the conventional method (POC$^{\text{es.}} = \sum n_i A_{\text{car.}}^{\text{conv.}} \, d_i^{B_{\text{car.}}^{\text{conv.}}}$) to the UVP-based particle profile under consideration to estimate the POC flux. The superscript "conv." distinguishes the parameters of the conventional method from those of the new approach. Using the globally optimized parameters reported by Guidi et al. (2008) ($A_{\text{car.}}^{\text{conv.}}$ = 12.5 and $B_{\text{car.}}^{\text{conv.}}$ = 3.81, for particle size range of 250 µm to 1.5 mm, with $n_i$ and $d_i$ in # L$^{-1}$ and mm, respectively), the conventional method underestimated the

measured fluxes by approximately a factor of 10 (data not shown). However, for the particle size range considered in this study, we found that the performance of the conventional method could be substantially improved by locally optimizing the fitting parameters, as also noted by Iversen et al. (2010). The locally optimized values were found to be $A_{\text{car.}}^{\text{conv.}}$ = 5.5237 and $B_{\text{car.}}^{\text{conv.}}$ = 1.1316. With the local fitting parameters, the conventional method also provided good estimates of POC flux measurement (Fig. 8a–f). Nonetheless, the obtained value for $B_{\text{car.}}^{\text{conv.}}$ is notably low leading to a strong decrease of the

contribution of large particles in the flux estimation at stations where large particles were present, such as in the deep core of the Sal eddy (Fig. 8b). This issue will be addressed further in the Discussion section. Locally optimizing the conventional method's PON flux parameters resulted in $A_{\text{nit.}}^{\text{conv.}}$ = 0.8926 and $B_{\text{nit.}}^{\text{conv.}}$ = 1.2354. However, since the associated error function did not show a global minimum for the expected range of parameters, these values were derived using a special adjustment (see Fig. A2 for details). With this adjustment, the conventional method also yielded good estimates of PON flux

measurements (Fig. 8g–l) and the corresponding molar C:N obtained from the conventional method (Fig. 8m–r) resulted in relatively good matches to the molar C:N directly measured from the DST-collected material. Despite this alignment with the measured values, the depth-specific estimated C:N, unlike the estimates from the new approach, remained unexpectedly almost constant at around 8.9 and did not vary with increasing depth.

### 3.5 POC, PON fluxes, and C:N fields along examined transects within the eddy


Depth-specific POC and PON fluxes, along with their corresponding C:N, were computed by applying the optimized parameters from presented approach to all UVP profiles conducted during the M160 expedition, which spanned four distinct transects (Fig. 2). The first transect traversed the sub-mesoscale frontal zone on November 27-28, 2019, followed by two transects across the Sal eddy, surveyed from November 29 to December 3, 2019. The fourth transect was conducted along

the Brava eddy-like event from December 12 to 17, 2019. Figure 9a–d depicts the position of considered stations along these




transects. The rim position of the Sal eddy during the surveys of each transect is shown in Fig. 9b and 9c for near-surface waters and at a depth of approximately 400 m. For each case, the estimated POC and PON fluxes, along with their corresponding C:N, were linearly interpolated and visualized across the examined transects as a function of water depth (Fig. 9e–h for POC flux, Fig. 9i–l for PON flux, and Fig. 9m–p for C:N). While an overall exponential decay of fluxes with

increasing depth is apparent, no distinct pattern emerges in the downward POC (Fig. 9e) and PON (Fig. 9i) fluxes along the transect within the frontal zone. Notably, along the transects through the Sal eddy, especially along the east-west transect, a distinct funnel-like pattern can be recognized for both POC and PON fluxes (Fig. 9g, k), showing approximately a two-fold increase in the fluxes within the eddy core at deep depths (around 1000 m) as compared to those at the same depths but outside of the eddy core. The influence of the Brava event resulted in a skewed funnel-like pattern in both the POC and PON

flux transects (Fig. 9h, l), primarily noticeable in the upper water column (< 500 m), where the POC and PON fluxes were elevated. This effect, however, diminished at greater depths. Overall, the particulate organic molar C:N increase from approximately 5.5 near the surface to about 12 at 1000 m depth. Upon comparison of all the C:N transects (Fig. 9m–p), noticeable variations are identified in the isolines of C:N within the Sal eddy: The C:N isolines deviates inwardly, suggesting lower C:N at deep depths within the core of the Sal eddy.

**3.6 Carbon degradation with respect to size of settling particle**

The size bin-specific carbon degradation rate ($\lambda_{ref.}/A_{car.}d^{B_{car.}}$) and remineralization length scale ($\lambda_{ref.}/(A_{car.}d^{B_{car.}} \times W)$ for surface water particles ($\ell = 0$) are shown in Fig. 10a and b. The bin-specific POC flux loss normalized by their surface water POC values ($\lambda \times t/(A_{car.}d^{B_{car.}})$), considering the effects of temperature and oxygen concentration variations on the degradation across various water depths is shown in Fig. 10c. As seen, the bin-specific degradation rate of POC slightly

increases with the size bin. Conversely, the remineralization length scale decreases as size bin increases. The POC flux loss associated with smaller size bins becomes considerable at greater depths, while for larger size bins, this effect is negligible at shallower depths. In Fig. 10e and f we have compared the estimated POC fluxes calculated from the new approach with the conventional method at two representative stations: one at Cape Verde Ocean Observatory (CVOO) station (see Fig. 2 for the position of CVOO), where the abundance of large particles was relatively lower, and one within the core of Sal eddy,

where large particles were more prevalent at the time of sampling (Fig. A3). To illustrate the impact of correcting for suspended particles in the same plots, we also applied our approach by setting $\lambda = 0$ (i.e., corrected for suspended particles but without the carbon degradation) and optimizing the fitting parameters. With this, the fitting parameters were obtained as $A_{car.}^0 = 1.0668\times10^{-3}$ and $B_{car.}^0 = 1.9525$, where the superscript "0" indicates that the degradation rate was set to zero for optimizing these parameters. Results show that at the CVOO station, the difference in profiles remains minimal down to

approximately 600–700 m, but gradually increases with increasing depth. At depths of 3500 m, the estimated POC flux by the conventional method and the new approach without degradation show an overestimation of roughly 100% compared to that estimated by the new approach with the degradation (Fig. 10e). In contrast, at the core of the Sal eddy, a station with a





high abundance of large particles, the conventional method significantly underestimates POC flux compared to the new

approach (Fig. 10f), although the impact of the degradation term in the new approach becomes less significant.

## 4 Discussion

### 4.1 Method I

Numerous studies, including Asper et al. (1992), Stukel et al. (2018) and Archibald et al. (2019), highlight the pivotal role of

sinking marine particles in the export of POM to the deep ocean. Using DSTs to sample sinking particles offers a direct

method for measuring POC and biogeochemical fluxes in the upper ocean layer. However, performing high-resolution

surveys of mesoscale eddies with DSTs poses technical challenges due to limited ship-time availability and the dynamic

nature of mesoscale eddies. Moreover, it requires labor-intensive sample processing, and subsequent laboratory analyses.

Consequently, often not more than one or two deployments are feasible when examining an eddy using DSTs during typical

cruise expeditions. On the other hand, ISCs offer more convenient operation and significantly shorter deployment times

(Kiko et al., 2017; Giering et al., 2020), making them valuable tools for conducting high-resolution surveys of size

distribution of particle and aggregate concentrations (abundance of aggregates and zooplankton) in the water column. They

are therefore particularly advantageous for observations within dynamic mesoscale and sub-mesoscale features in the ocean.

Using ISC systems, we can effectively assess the size distribution of particle concentrations along prespecified transects

across an eddy both horizontally and vertically (Waite et al., 2016). Given that ISC-derived depth-specific particle

concentrations are regarded as proxies of the abundance of sinking particles at corresponding depths, the ISC particle

profiles can be calibrated against corresponding sediment trap flux measurements in a region (where available) to provide

reliable regional estimates of POC (and other biogeochemical) fluxes in the water column (Iversen et al., 2010, Clements et

al., 2023). As such, when combined with DSTs, ISCs facilitate high-resolution spatial projection of biogeochemical flux

fields along surveyed transects within eddies. However, an inherent problem associated with using ISC-derived particle

concentrations as a basis for estimating particle flux lies in the inability to distinguish between sinking and suspended

particles in the ISC images (McDonnell and Buesseler, 2012; Cael and White, 2020). Despite fitting ISC particle

concentration data to the flux measurements for estimating biogeochemical fluxes, the uneven distribution of suspended

particles might introduce inaccuracies in ISC-based estimations of biogeochemical fluxes in the water column. This issue

can become more pronounced within mesoscale eddies, as they locally displace isopycnals over relatively short timescales

(Sweeney, 2002), thereby exacerbating the uneven distribution of suspended particles. Furthermore, the exclusion of

zooplankton from ISC particle data is also a time intensive task (Giering et al., 2020). The first method introduced in this

study endeavors to tackle these issues. By combining gel trap-based SDPF with their converted ISC counterparts derived

from Eq. 1, we have introduced a novel method that accounts for the contribution of non-settling particles (suspended

particles and zooplankton) in the converted profiles of ISC-based SDPF. Overall, when non-settling particles are included,





the fitted lines characterizing the log-transformation of ISC-based SDPF exhibit notably higher intercepts compared to their

gel trap-based counterparts, alongside slightly steeper slopes (Fig. 7a, c), primarily due to high concentration of small suspended particles. To address this, we applied the correction factors α and β to adjust the intercept and slope of the characterizing fitted lines accordingly (Fig. 7b, d). Suspended particles wield a substantial influence on ocean particle dynamics by either being a product of disaggregation (Briggs, 2020) or fostering aggregation through increased collision rates (Kiørboe, 2001). Although we used Method I to provide reliable estimates of the SDPF as an essential input for

accurate POC and PON flux estimations in Method II (see Fig. 1), this method has the potential to offer insight into the suspended particle concentration in the water column. For example, we estimated that at 100 m depth the average concentration of suspended particles in the 100-160 µm size range was approximately 1.2-fold higher than the abundance of sinking particles of the same size at the stations where DSTs were used. This inference, however, leans on the assumption that suspended particles tend to be considerably smaller in size compared to swimmers, which were typically measured to be

larger than 0.2 mm.

## 4.2 Method II

The equation for estimating POC flux in the conventional method ($POC^{es.} = \sum n_i A^{conv.}_{car.} d_i^{B^{conv.}_{car.}}$) assumes that individual particles of the same size contain an equal amount of organic carbon irrespective of their water depth and that the carbon flux attenuation in the water column is solely from a decline in particle numbers with depth. However, depending on type and

composition, some particles primarily composed of organic matter may retain a nearly constant apparent size (ISC-detectable) while their organic carbon content decreases due to microbial degradation as they age. Some other particles, especially those with substantial inorganic components like zooplankton shells, lithogenic minerals, or diatom frustules, may retain their apparent size but lose most or all of their organic carbon at certain depths. (see Fig. A4 for a schematic representation of these scenarios). In instances where ISC particle data are fitted with flux measurements from both the

upper and lower parts of the water column—obtained from drifting and moored sediment traps, respectively—the optimized parameters $A^{conv.}_{car.}$ and $B^{conv.}_{car.}$ characterize the estimated POC flux associated with particles from both fresh and older material, effectively interpolating between the POC flux of particles from different water depths.  However, due to the dynamic nature of eddies and associated sampling limitations, POC flux measurements using DST from deeper depths beneath the eddies are often unavailable or unreliable. Consequently, estimating POC flux with ISC particle profiles at

deeper depths may lack accuracy because of the absence of flux measurements to correlate to. Similar issues arise for shallower depths when an eddy passes a moored sediment trap, leaving only deep flux measurements available (Fischer et al., 2021). These considerations also apply to estimating PON flux. To address this, we used the size-velocity relationship of in-situ collected aggregates to approximate particle age at a specified water depth, acknowledging that particles degrade with age. Since sinking velocity controls the travel time of particles from the surface to a given depth, it directly controls the age

of settling particulate matter. The particle age was estimated as the travel time from the surface water to the target water




depth $\ell$, determined by the particle sinking velocity associated with the particle size, $t = \ell/W$. Additionally, to account for the reduction in carbon degradation rates with decreasing water temperature and potentially with decreasing oxygen concentration, we respectively applied a $Q_{10}$ formulation and Michaelis–Menten kinetics using previously optimized values of $Q_{10} = 2.5$ and $K_m = 19$ µmol $O_2$ kg$^{-1}$ (DeVries and Weber, 2017). These values align with those reported in Iversen and

Ploug (2013) and Ploug and Bergkvist (2015) for diatom aggregates measured on roller-tank made diatom aggregates. The inclusion of Michaelis–Menten kinetics is particularly useful for better understanding POC export in anoxic eddies forming in the ETNA (Fiedler et al., 2016). Note that, by setting $Q_{10} = 1$ and $K_m = 0$, the degradation term can be made independent of temperature and oxygen concentration. The improved formulation for estimating POC flux (Eq. 5) integrates critical factors that influence the export of POC to the deep ocean, namely particle degradation rate and sinking velocity, and

demonstrates a good alignment with the DST-based measured POC fluxes (Fig. 8a–f). Moreover, it facilitates the examination of the individual effects of these factors on POC flux and offers the flexibility to incorporate additional factors in the future.

Typically, the C:N of the fresh bulk marine POM (near surface) often range between 4 and 6, and it is anticipated to increase with increasing depth as sinking particles age and degrade (Romankevich, 1984; Schneider et al., 2003), a trend also

observed in the DST-based measured C:N (Fig. 6c). However, the estimated C:N from the conventional method tends to remain quasi constant with increasing depth. To resolve this, we reformulated the estimated PON flux in Method II (Eq. 7) by assuming a correlation between the organic nitrogen and organic carbon content in sinking particles, as evidenced by Fig. 6d. This assumption implicitly accounts for preferential degradation of nitrogen by microbes associated with sinking particles (Grossart and Ploug, 2001). However, given that organic nitrogen degrades more rapidly than organic carbon within

sinking aggregates (Iversen, 2023), we introduced a particle age-related correlation factor ($A_{\text{nit.}}\, t^{A_{\text{nit.}}}$) in which the particle age at a given depth is estimated as the particle travel time from surface to that depth ($t$) using the size-velocity relationship. As shown in Fig. 8g–r, the new parameterization for PON flux not only produced estimates that are in good agreement with the PON measurements from the DSTs, but also captured the general decrease of C:N with increasing water depth (Fig. 6c). This trend is consistent with observations of PON flux documented for the deep waters of the Cape Verde region at the

CVOO station (Fischer et al., 2021). As such, correlating PON with POC and particle age, specified by particle sinking velocity, seems to be a promising approach for providing more reliable estimates of PON flux in the water column using in-situ optics, especially within dynamic features such as eddies.

## 4.3 Flux fields across the examined transects

As illustrated, the Sal eddy showed a distinct funnel-like pattern down to approximately 1000 m, particularly along the east-west transect. One possible scenario explaining this pattern is the formation and accumulation of larger aggregates at the rim of the eddy near the surface due to increased concentration of phytoplankton cells and higher shear flow. In response to the circular flow field within the upper part of the eddy, these particles are partially drawn into the eddy's core as they settle



(Waite et al., 2016), leading to nearly a two-fold flux enhancement at the deep core around 1000 m compared to the same
depths outside the eddy core. Eddy-associated flux enhancements in this region have been reported previously (Fiedler et al., 2016, Fischer et al., 2016, 2021), mainly associated with oxygen-depleted mode-water anticyclonic eddies. Nevertheless, although the Sal eddy was a relatively weak cyclone and not oxygen-limited, it still had significant flux enhancement at its deep core. Above the deep enhanced flux, the Sal eddy exhibited a shallower part (at depths of approximately 100–400 m) with lower POC flux (Fig. 9g), possibly associated with upwelling and elevated pycnoclines at its shallow core, typical for
cyclonic eddies (McGillicuddy 2016 and 2007). The skewed funnel-like pattern in POC flux at depth in the Brava event was less pronounced and relatively shallow (Fig. 9h). This might be attributed to the stage of this event as it was in an unsteady state at the time of sampling, so that any observed pattern may have been spuriously created, or expected patterns may have been destroyed. We observed similar PON flux fields to those observed for POC, reflecting the correlation between the organic nitrogen and carbon contents of sinking particles, which we took into account in the formulation of the estimated
PON flux (Eq. 7). Still, comparing the C:N fields, we found lower C:N at greater depths in the core of the Sal eddy, suggesting that the sinking particles at depth were fresher (less degraded) at the eddy core compared to the surrounding waters at similar depths. A plausible explanation for this observation is the higher prevalence of faster-sinking aggregates in the eddy core at deeper depths, as organic matter in particles with faster sinking speeds is generally supposed to be fresher when considering a specific water depth. Since, typically, larger aggregates tend to sink faster than smaller ones (Fig. 5), this
suggests that the observed lower C:N at the deep core of the Sal eddy can be attributed to an increased concentration of larger, faster-settling aggregates in the deep core of the eddy, potentially formed only a few days earlier at the eddy rim near the surface and sunk downward along the salinity isolines in a funnel-like movement, or they were partially drawn toward the eddy core by the circular flow field in the upper part of the eddy as they sank out of the surface water.

## 4.4 Significance of particle size and suspended particles

The increase in bin-specific degradation rate of POC with size bin (Fig. 10a) can be attributed to the higher porosity of larger particles, which facilitates carbon degradation (Alldredge and Gotschalk, 1998; Ploug and Passow 2007). Conversely, the decrease in remineralization length scale with size bin (Fig. 10b) suggests that smaller particles are fully degraded at shallower depths compared to larger ones. Despite the relatively strong decrease in carbon degradation rate with decreasing temperature in the water column ($Q_{10} = 2.5$), the bin-specific POC flux loss becomes significant at deep depths, especially for
small bins. This is because smaller particles tend to sink more slowly than larger particles. For instance, based on the measured size-velocity relationship (Fig. 5), a 0.2 mm aggregate takes approximately 40 days to reach a depth of 2000 m, while a 2.0 mm aggregate takes only about 12 days. The discrepancy between POC flux estimates from our approach and the conventional method (Fig. 10e, f) is largely due to the relatively low value of $B_{car.}^{conv.} = 1.13$, which reduces the contribution from larger particles and allocates more carbon flux to smaller size bins. As shown in Fig. 8a–f, this leads the conventional
method to estimate significantly lower POC fluxes at depths shallower than 100 m compared to the new approach, given the high abundance of large particles near the surface. Applying this low $B_{car.}^{conv.}$ value along Sal eddy transects diminishes the



observed funneling effect (Fig. A5) due to the relatively higher concentration of large particles within the eddy core, particularly at depth. It is important to note that when comparing $B_{\text{car.}}^{\text{conv.}}=1.13$ with the new approach's $B_{\text{car.}}=1.92$, the latter requires an adjustment by adding 0.5. This adjustment accounts for the separation of sinking velocity ($B_w=0.5$) and carbon

content in the POC flux estimation under the new approach (see Table A1 for parameter dimensions in both approaches). A plausible explanation for the conventional method yielding such a low value for $B_{\text{car.}}^{\text{conv.}}$ is the relatively small lower bound of the particle size range considered (0.064–4.1 mm). This could result in the smallest size bins encompassing a substantial number of suspended particles (as previously exemplified), thereby artificially increasing their contribution to the estimated flux. Consequently, the optimization algorithm yields a low $B_{\text{car.}}^{\text{conv.}}$ to appropriately balance their contribution in the

estimated flux. Another reason that could artificially increase the contribution of small particles to the estimated flux is that the conventional method does not account for the size-velocity relationship. Since smaller particles tend to sink more slowly, incorporating a typical size-velocity will downweigh the contribution of these particles in the flux calculation. To mitigate the risk of including many suspended particles using the conventional method, one approach is to consider a larger minimum size bin. However, this adjustment could pose challenges, particularly in areas where POC flux is primarily driven by small

(and slow sinking) particles, as suggested in several studies (Dall'Olmo and Mork, 2014; Omand et al., 2015; Baker et al., 2017). In this study, we set the minimum size bin to 64–80 µm, as it was the smallest size bin for which the UVP5 provided data, and the maximum size bin to 3.25–4.1 mm, reflecting the largest sinking particles collected by the gel traps in the study area.

**5. Conclusion**

Our methodological developments, building on the pioneering work of Guidi et al. (2008), devised a novel mechanistic approach for estimating ISC-based POC and PON fluxes by directly incorporating size-specific sinking velocities and carbon degradation rates of settling particles. This approach minimizes the contribution of non-settling particles in ISC-based biogeochemical flux estimates and accounts for variations in carbon degradation rates driven by changes in water temperature and oxygen concentration at different depths. By incorporating locally measured size-velocity relationship and

degradation rates of settling particles, it enables reliable, high-resolution estimations of spatiotemporal POC and PON fluxes and their variability throughout the water column. This capability makes the approach particularly valuable for quantifying export fluxes in sub- and meso-scale dynamic features. When integrated into biogeochemical models, it helps to better understand the impact from meso-scale features, such as eddies, to total carbon export and the efficiency of the biological carbon pump (Turner, 2015). With the continuously growing numbers of ISC-based particle profiles across the global ocean

(Kiko et al., 2022), this approach has the potential to improve the accuracy of carbon export and sequestration rates (Lombard et al., 2019; Siegel et al., 2016; Giering et al., 2017), an estimate which large-scale models cannot agree on at the moment (Henson et al., 2022).





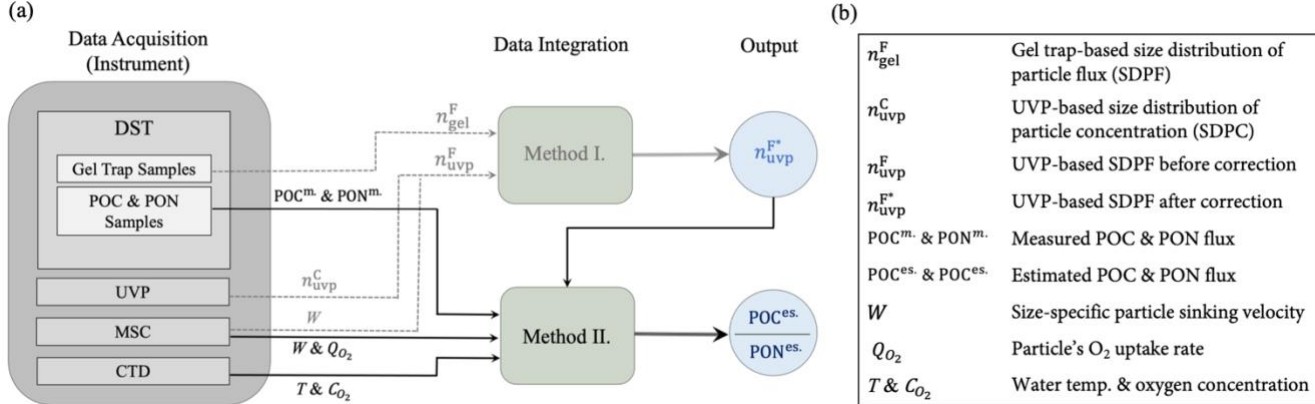


**Figure 1.** (a) A schematic diagram depicting the input/output processes of the two introduced methods for integration of in-situ observations and their interconnections (see Materials and Methods). (b) Definition of the primary notations used in the methodology. The abbreviations DST, UVP, MSC, and CTD refer to free-drifting sediment traps, Underwater Vision Profiler, Marine Snow Catcher, and CTD-Rosette system, respectively.






**Figure 2.** (a) Map of station locations in the ETNA, within the Cape Verde archipelago region, sampled during the M160 expedition (Nov.–Dec. 2019). Arrows indicate the dominant flow directions of major current systems in the area: Canary Current (CC), North Equatorial Current (NEC), Mauritanian Current (MC), Cape Verde Current (CVC), and Poleward Undercurrent (PUC). The depiction of primary current systems is adapted from Fig. 1 in Romero and Ramondenc (2022). CVOO marks the position of the Cape Verde Ocean Observatory. (b–d) Enlarged views of stations within the three main sampling zones. In zones A1, A2, and A3, we sampled a frontal zone, a cyclonic eddy (referred to as the Sal eddy), and a shallow, elliptical-shaped eddy-like feature (referred to as the Brava eddy-like event). The chlorophyll maps represent the average satellite-derived chlorophyll *a* concentration (cds.climate.copernicus.eu) for the sampling period within each zone: A1 (25/11–01/12/2019), A2 (25/11–03/12/2019), and A3 (10–19/12/2019). The satellite-derived surface currents are shown for the midpoint of each period.



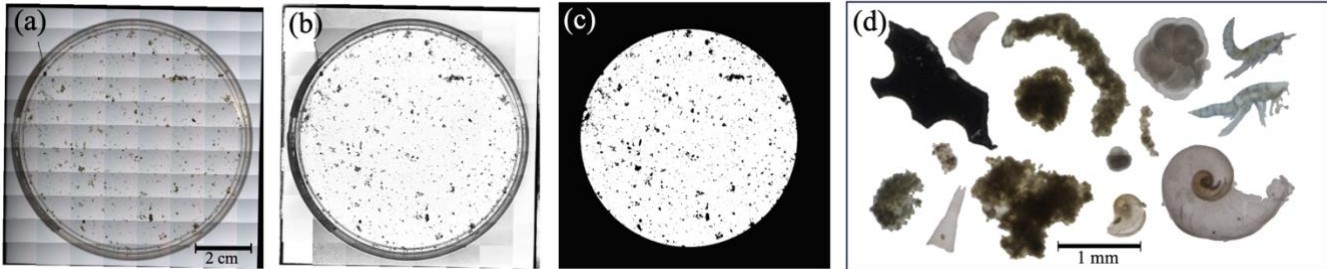

**Figure 3.** Image analysis of gel trap sample using Gel-PISA. (a) High-resolution reconstruction of the entire gel trap cup image, achieved by stacking and stitching subsection frames. (b) Conversion of the final image to grayscale, followed by background removal. (c) Transformation of the grayscale image into a binary format for particle counting and size calculation. (d) Examples of particle images extracted from the gel trap sample image.




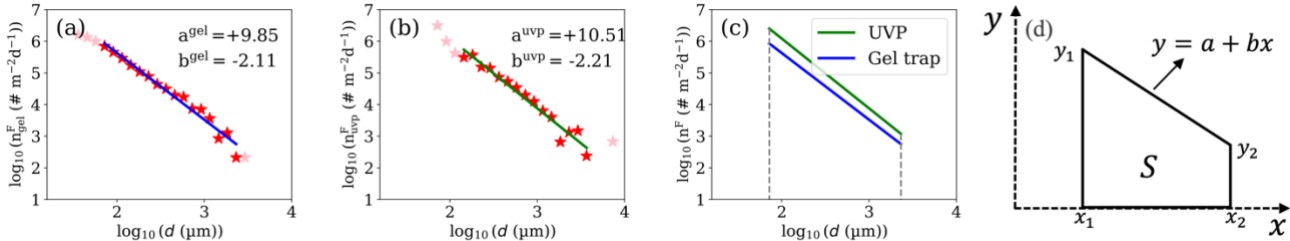

**Figure 4.** Characterization of particle flux size distribution. (a) Example of a particle flux size distribution characterized using the best linear fit for a typical gel trap sample. (b) Corresponding UVP-based particle flux size distribution characterized in the same manner. (c) Comparison of the characterized lines from panels (a) and (b) for the particle size range of 0.64–4.1 mm. (d) Geometric representation of a particle flux size distribution, depicted as a right trapezoid.



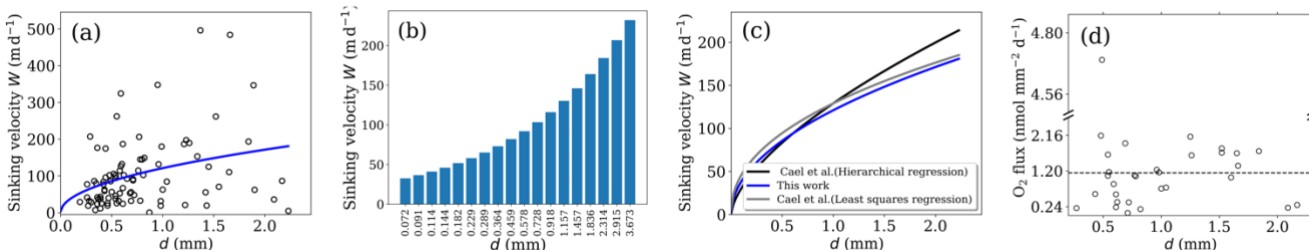

**Figure 5.** (a) Measured particle sinking velocity $W$ versus particle size $d$ (equivalent spherical diameter). The blue curve represents the best fit of the power function $W(d) = A_w d^{B_w}$, with $A_w$=120.9917 and $B_w$=0.5005 ($R^2$=0.11). (b) Sinking velocities corresponding to the considered size bins, derived from the size-velocity relationship. The smallest size bin (64–80 µm) corresponds to the minimum particle sizes detectable by the UVP5 camera, while the largest bin (3.25–4.1 mm) represents the typical size of largest sinking particles captured by the gel trap. (c) Comparison of size-velocity relationships for sinking marine particles from this study and those in Cael et al. (2021). (d) Calculated diffusive $O_2$ flux across the interface of collected aggregates versus particle size, with the dashed line indicating the mean flux obtained.



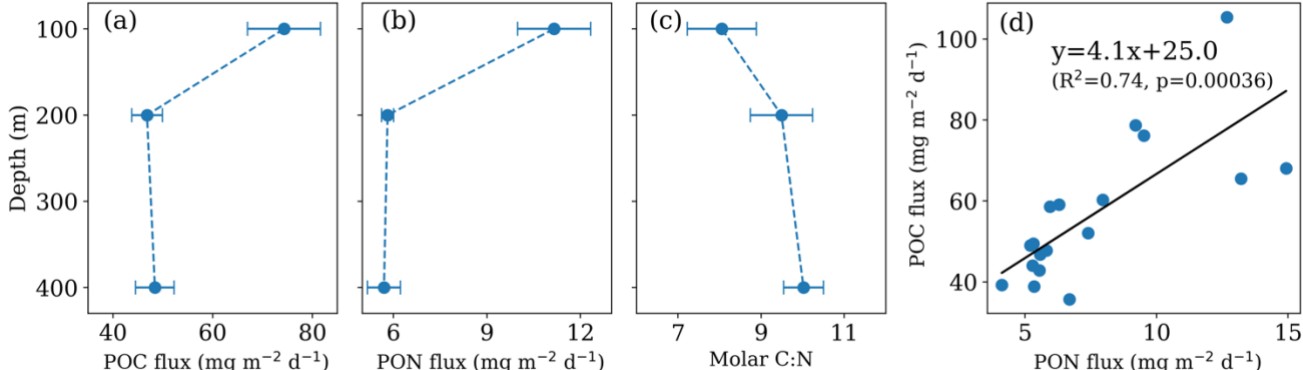

**Figure 6.** Mean DST-based fluxes (N=6): (a) POC flux, (b) PON flux, (c) Corresponding molar C:N calculated as 14/12×(POC/PON). (d) Individual POC fluxes vs. their corresponding PON fluxes. The black regression line shows the correlation between measured POC and PON fluxes.





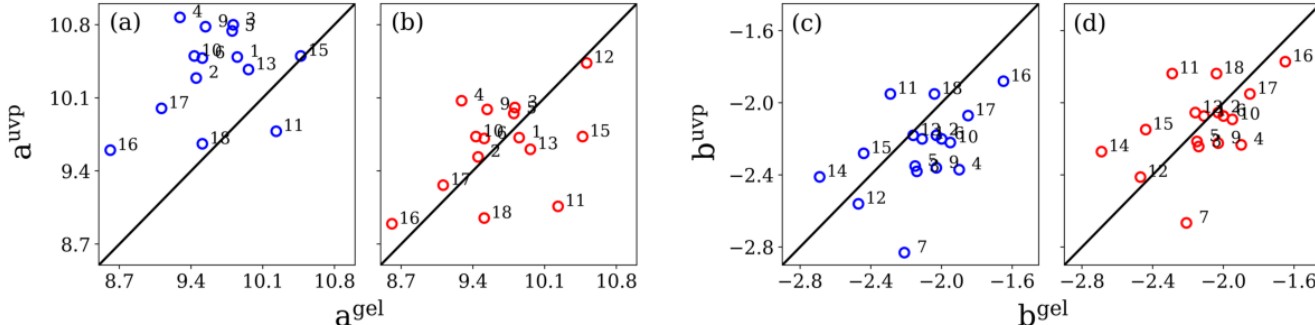

**Figure 7.** The intercepts and slopes of the characteristic fitted lines for the log-transformed gel trap-based SDPF $(\log_{10}(n_{gel}^F))$ are compared with those of their UVP-based counterparts $(\log_{10}(n_{uvp}^F))$. Panels (a) and (c) display the UVP intercepts and slopes before correction (blue circles), while panels (b) and (d) show the results after correction (red circles). The numbers on the circles correspond to the sample index "k" (see Table A2), with the black lines indicating the 1:1 line.





**Figure 8.** Estimation of POC flux, PON flux, and corresponding molar C:N versus water depth: Panels (a–f) show estimated POC flux, (g–l) PON flux, and (m–r) the corresponding molar C:N at the six DST deployment stations. For each case, fluxes were calculated using the average of two UVP particle profiles: one corresponding to the deployment and one to the recovery of the respective DST. The red stars represent DST-based measurements, while the black and grey curves correspond to estimates using the new approach and the conventional method with local parameter optimization, respectively. Errors indicate the relative deviations of the current approach estimates from the measurements.





**Figure 9.** Station locations in the examined transects: Panels (a), (b), and (c) show the locations of stations through the frontal zone, the Sal eddy, and the Brave eddy-like event, respectively, as depicted on the map of the study area (Fig. 2). In panels (a) and (b), red and blue circles represent the rim of the Sal eddy at the surface and at a depth of 400 m, respectively. Estimated fluxes using the presented approach: Panels (e–h) show POC flux, (i–l) PON flux, and (m–p) the corresponding C:N versus water depth along all the examined transects.





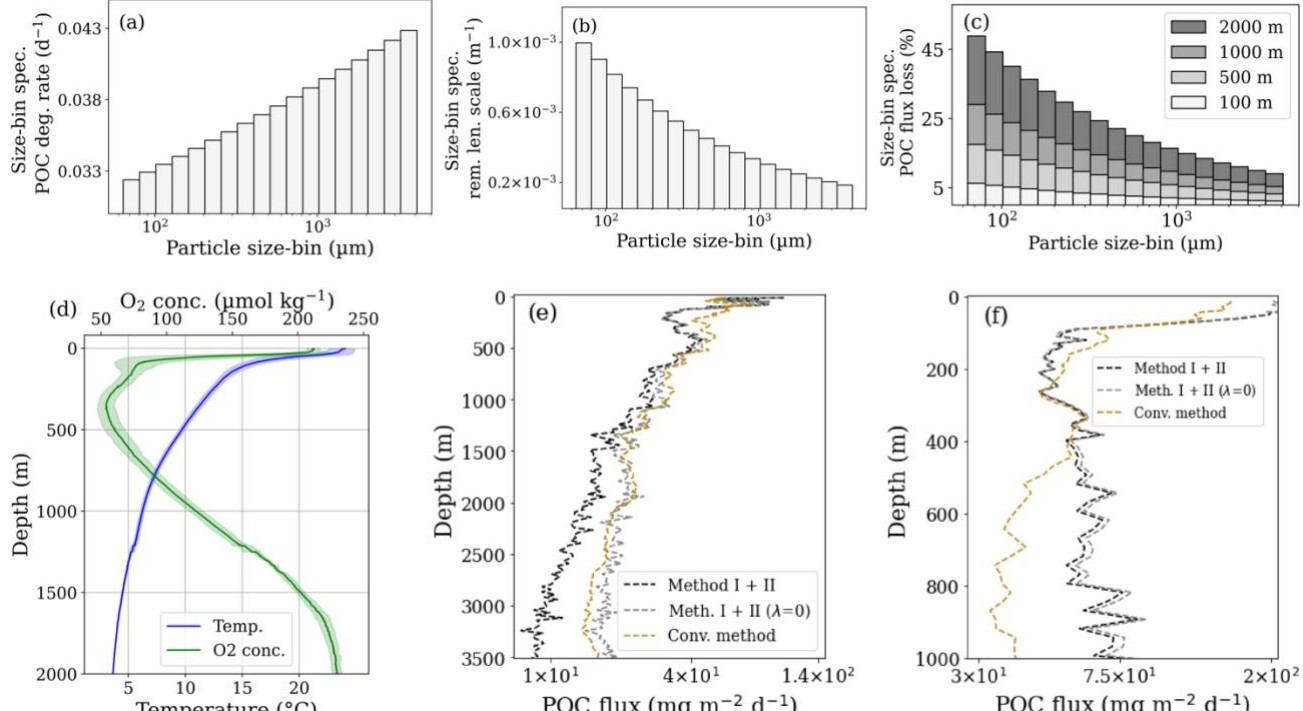

**Figure 10.** (a) Size-bin-specific POC degradation rates. (b) Remineralization length scale for surface water particles, excluding the effects of temperature and O₂ concentration variation. (c) POC flux loss, normalized by their surface values ($A_{car.}d^{B_{car.}}$) for each size bin, accounting for temperature and oxygen effects. (d) Average of all (N=73) measured temperature and oxygen profiles in the study area used for the calculation of bin-specific POC flux loss shown in panel (c). (e, f) Comparison of estimated POC fluxes using the new approach and the conventional method at the CVOO station (e) and at the core of the Sal eddy (f): dark grey (new approach, corrected for the contribution of suspended particles and including the degradation term), light grey (new approach without the degradation term ($\lambda = 0$), but corrected for the contribution of suspended particles), light brown (conventional method after local optimization of the associated parameters). See Fig. A3 for a comparison of abundance and total particle volume of small (0.064 mm < d < 0.5 mm) and large particles (0.5 mm < d < 4.1 mm) at the core of Sal eddy and CVOO station.



**Appendix A: Supplementary plots and tables**

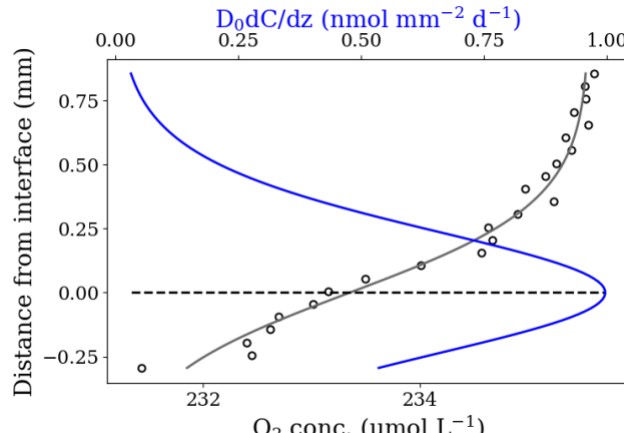

**Fig. A1.** Oxygen profile across the water-aggregate interface and corresponding diffusive oxygen flux. The data points, represented by circular markers, show a typical oxygen profile measured on board using the microsensor through the water-

675 aggregate interface of an in-situ collected aggregate in the flow chamber system during the M160 expedition. The black curve depicts the modelled oxygen profile, while the blue curve shows the corresponding diffusive flux, calculated using Fick's first law as described in Moradi et al. (2021). The diffusion coefficient of oxygen ($D_0$) was adjusted based on the in-situ temperature and salinity of the water from which the aggregate was collected. The dashed line marks the position of the water-aggregate interface in the measured profile.

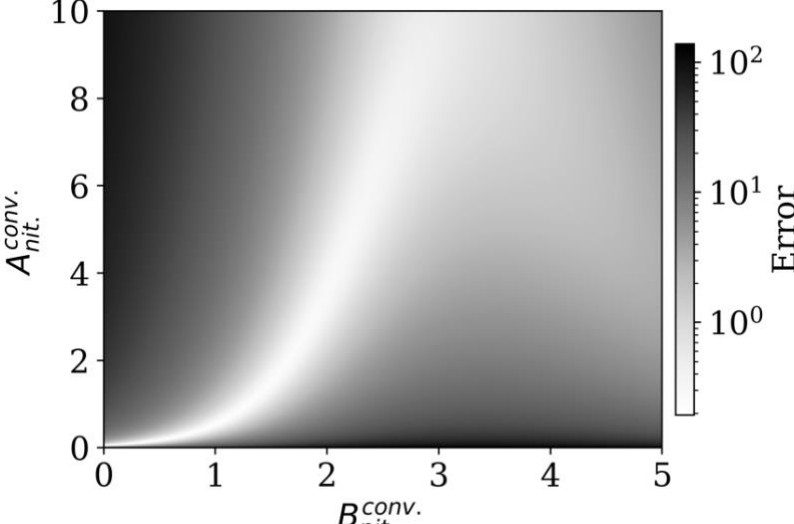

**Figure A2.** Visualization of the error landscape for estimating PON flux using the conventional method as a function of optimization parameters $A_{\mathrm{nit.}}^{\mathrm{conv.}}$ and $B_{\mathrm{nit.}}^{\mathrm{conv.}}$. The error was calculated as Error $= \sum_k [\log_{10}(\mathrm{PON}_k^{\mathrm{m.}}) - \log_{10}(\mathrm{PON}_k^{\mathrm{es.}})]^2$, where $\mathrm{PON}_k^{\mathrm{m.}}$ are DST-based PON flux measurements, and $\mathrm{PON}^{es.} = \sum_i n_i A_{\mathrm{nit.}}^{\mathrm{conv.}} d_i^{B_{\mathrm{nit.}}^{\mathrm{conv.}}}$ summing over all size bins within the size range of 64 µm to 4.1 mm. The error landscape reveals that there are no local or global minimum 'within' the expected ranges of $A_{\mathrm{nit.}}^{\mathrm{conv.}}$ and $B_{\mathrm{nit.}}^{\mathrm{conv.}}$, with the error consistently decreasing as both parameters become smaller. To circumvent this issue, a workaround is to fix $A_{\mathrm{nit.}}^{\mathrm{conv.}}$ to 2.02/12.5 and optimize only $B_{\mathrm{nit.}}^{\mathrm{conv.}}$. The factor 2.02/12.5 is the ratio of globally optimized values of $A_{\mathrm{nit.}}^{\mathrm{conv.}}$ and $B_{\mathrm{nit.}}^{\mathrm{conv.}}$ reported in Guidi et al. 2008.





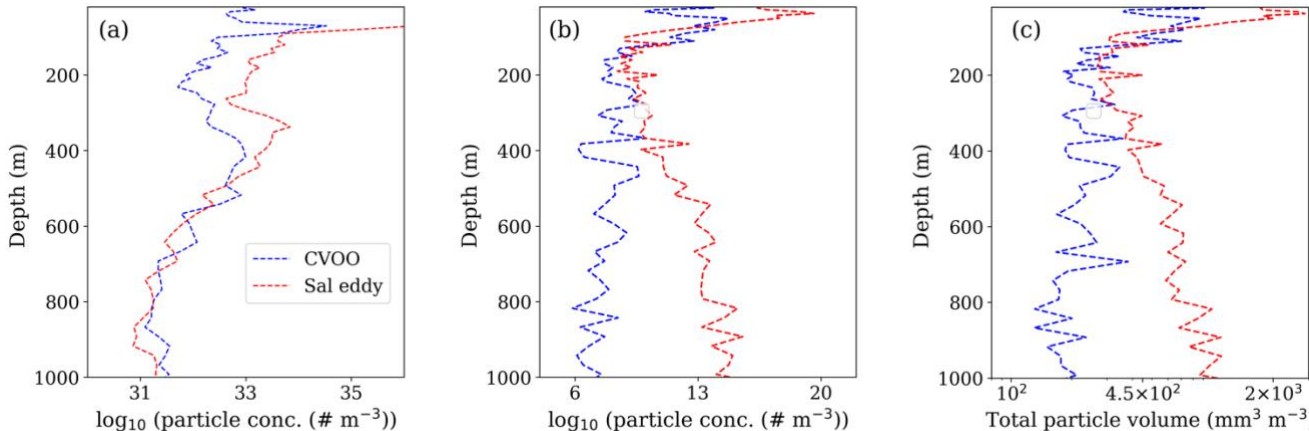

**Figure A3.** Particle concentrations detected by UVP within the size range of 0.064 – 4.100 mm are compared between the core of Sal Eddy and the CVOO station. Panels (a) and (b) show UVP-based concentrations for small (< 0.5 mm) and large (> 0.5 mm) particles, respectively. Particle concentrations for size bins within these sub-ranges were first log-transformed and then summed. Panel (c) shows the total volume of particles detected, expressed in mm³ per cubic meter of seawater.





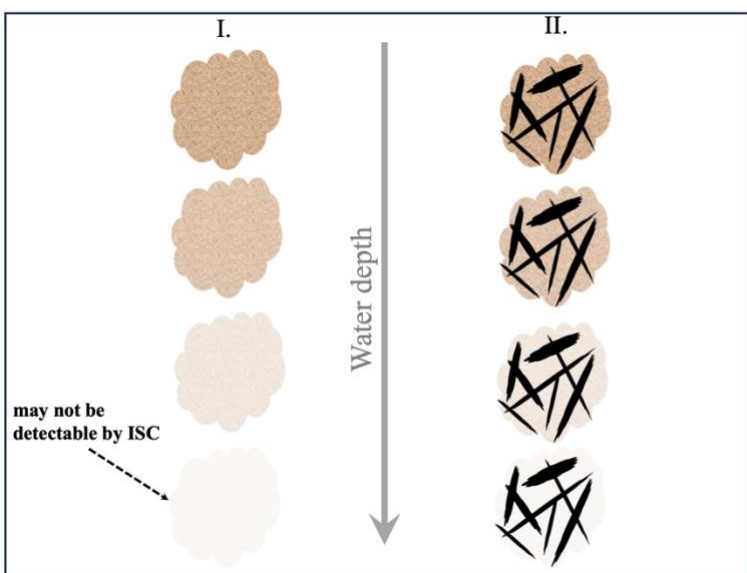

**Figure A4.** A speculative representation of potential issues in estimating POC flux using the conventional method. This schematic illustrates the potential problems in estimating POC flux based on ISC-detected particle concentrations, under the assumption that carbon content is size-dependent but not depth-dependent, as is the case in the conventional method.

Scenario I.: The particle is primarily composed of organic carbon. As it sinks, microbial degradation reduces its organic carbon content, while its apparent size, detectable by ISC, remains mostly unchanged. However, as the particle becomes increasingly porous and degraded, it may eventually become undetectable by ISC. Scenario II.: The particle has a significant inorganic structure. Despite a decrease in its organic carbon content with depth, its apparent size remains mostly constant, even when it loses all its organic carbon.



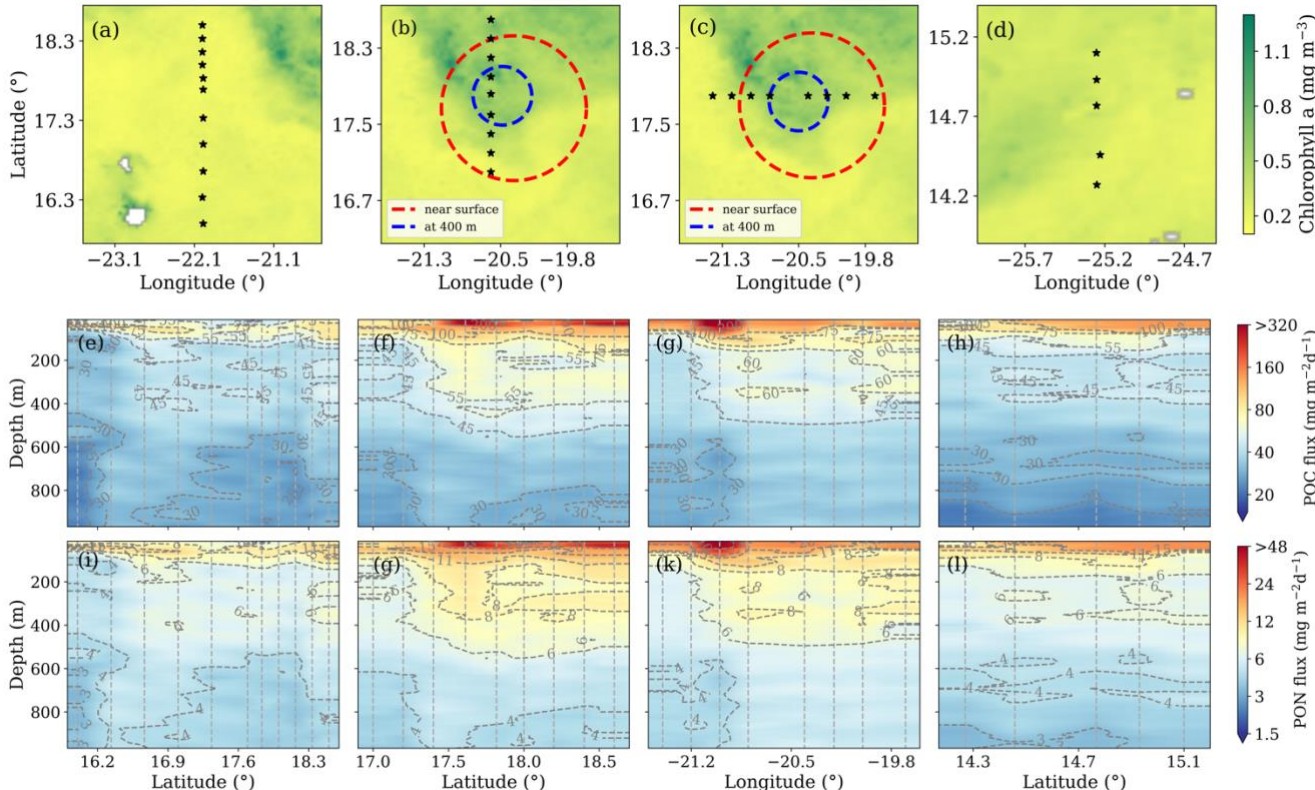

**Figure A5.** Fields of POC and PON fluxes along the examined transects during M160 expedition using the conventional method. Panels (a–d) show the examined transects, while panels (e–h) and (i–l) display the POC and PON flux fields, respectively, along these transects. The fluxes were obtained using the conventional method, with parameters locally optimized for a particle size range of 0.064 – 4.100 mm. The dashed circles in panels (b) and (c) indicate the position of the Sal eddy. Refer to Fig. 9 for a comparison with results obtained using the new approach.



**Table A1.** Dimension of parameters defined in this study. List of parameters, their descriptions, and corresponding dimensions. The dimensions are denoted as M for mass, $L$ for length, T for time, and Θ for temperature.

| Parameter | Description | Dimension |
|---|---|---|
| $W$ | Particle sinking velocity | $LT^{-1}$ |
| $d$ | Particle size | $L$ |
| $A_w$ | Coefficient of particle size-velocity relationship ($W = A_w d^{B_w}$) | $L^{1-B_w}T^{-1}$ |
| $B_w$ | Exponent of particle size-velocity relationship ($W = A_w d^{B_w}$) | - |
| $D_i, D_{i+1}$ | Lower and upper bounds of $i^{th}$ size bin | $L$ |
| $n_{uvp}^{C}$ | UVP-based size distribution of particle concentration | $L^{-3}$ |
| $n_{gel}^{F}, n_{uvp}^{F}$ | DST-based and UVP-based size distribution of particle flux | $L^{-2}T^{-1}$ |
| $D_i, D_{i+1}$ | Lower and upper bounds of $i^{th}$ size bin | $L$ |
| $n_{uvp}^{C}$ | UVP-based size distribution of particle concentration | $L^{-3}$ |
| $n_{gel}^{F}, n_{uvp}^{F}$ | DST-based and UVP-based size distribution of particle flux | $L^{-2}T^{-1}$ |
| $a^{gel}, a^{uvp}$ | Intercept of the characteristic fitting line for $\log_{10} n_{gel}^{F}$ and $\log_{10} n_{uvp}^{F}$ vs $\log_{10} d$ | - |
| $b^{gel}, b^{uvp}$ | Slope of the characteristic fitting line for $\log_{10} n_{gel}^{F}$ and $\log_{10} n_{uvp}^{F}$ vs $\log_{10} d$ | - |
| $S^{gel}, S^{uvp}$ | Trapezoid area associated to the log-log transform of particle flux vs particle size for gel trap and UVP | - |
| $\alpha, \beta$ | Correction factors for $a^{uvp}$ and $b^{uvp}$, respectively, to exclude non-sinking particles | - |
| $n_{uvp}^{F^*}$ | Estimated UVP-based size distribution of particle flux after correction, $n_{uvp}^{F^*} = 10^{(\alpha \times a^{uvp})} d^{(\beta \times b^{uvp})}$ | $L^{-2}T^{-1}$ |
| $POC^{m.}, PON^{m.}$ | Trap-based measured POC and PON flux | $ML^{-2}T^{-1}$ |
| $C:N$ | Molar ratio of POC flux and PON flux | - |
| $M^{car.}$ | POC mass of the model particle ($M^{car.} = A_{car.} d^{B_{car.}} - \lambda \times t$) | $M$ |
| $A_{car.}$ | Coefficient of the function for the $M^{car.}$ | $ML^{-B_{car.}}$ |
| $B_{car.}$ | Exponent of the function for the $M^{car.}$ | - |
| $\lambda$ | Particle carbon degradation rate | $MT^{-1}$ |
| $\ell$ | Water depth at which the fluxes are calculated | $L$ |
| $t$ | Estimated particle travel time from surface to the considered depth $\ell$, $t = \ell/W$ | $T$ |
| $\lambda_{ref.}$ | Particle carbon degradation rate at a given water temperature and oxygen concentration | $MT^{-1}$ |
| $\lambda_0$ | Particle carbon degradation rate per particle surface area | $ML^{-2}T^{-1}$ |
| $J_{O_2}$ | Particle interfacial diffusive oxygen flux | $ML^{-2}T^{-1}$ |
| $Q_{O_2}$ | Oxygen uptake rate by the particle (aggregate) | $MT^{-1}$ |
| $O_{2,con.}$ | oxygen concentration of water | $ML^{-3}$ |
| $K_m$ | oxygen's half-saturation constant of particle | $ML^{-3}$ |
| $\Lambda$ | Particle surface area | $L^2$ |
| $\emptyset$ | Stoichiometric factor for converting oxygen to carbon | - |
| $\sigma$ | Molar mass of carbon | $M$ |
| $T$ | Water temperature | Θ |
| $T_{ref.}$ | Temperature at which the $\lambda_{ref.}$ has been measured | Θ |
| $M^{nit.}$ | Organic nitrogen mass of the model particle ($M^{nit.} = \gamma M^{car.} = A_{nit.} t^{B_{nit.}} M^{car.}$) | $M$ |
| $\gamma$ | Correlation factor between $M^{nit.}$ and $M^{car.}$ ($M^{nit.} = \gamma M^{car.}$) | - |
| $A_{nit.}$ | Coefficient of the function for the correlation factor ($\gamma = A_{nit.} t^{B_{nit.}}$) | $T^{-B_{nit.}}$ |
| $B_{nit.}$ | Exponent of the function for the correlation factor ($\gamma = A_{nit.} t^{B_{nit.}}$) | - |
| $A_{car.}^{conv.}$ | Coefficient of the function of the estimated POC flux in the conventional method | $ML^{-2-B_{car.}^{conv.}}T^{-1}$ |
| $B_{car.}^{conv.}$ | Exponent of the function of the estimated POC flux in the conventional method | - |
| $A_{nit.}^{conv.}$ | Coefficient of the function of the estimated PON flux in the conventional method | $ML^{-2-B_{nit.}^{conv.}}T^{-1}$ |
| $B_{nit.}^{conv.}$ | Exponent of the function of the estimated PON flux in the conventional method | - |



**Table A2.** Measured DST-based POC and PON fluxes with corresponding molar C:N, alongside the intercept and slope of the characteristic lines fitted to the log-transformed gel trap-based size distributions of particle flux and their UVP counterparts. Note that the values for $a^{\mathrm{uvp}}$ and $b^{\mathrm{uvp}}$ are presented without correction. Both particle flux ($n_{\mathrm{gel}}^{\mathrm{F}}$ and $n_{\mathrm{uvp}}^{\mathrm{F}}$) as well as particle size $d$ were initially normalized by $1\ \mathrm{m^{-2}d^{-1}}$ and $1\ \mu\mathrm{m}$ respectively, prior to log-transformation. Consequently, the intercepts and slopes are unitless.

| DST sample reference | | | | Measured Biogeochemical fluxes | | | Gel trap-based particle flux $\log_{10} n_{\mathrm{gel}}^{\mathrm{F}} = a^{\mathrm{gel}} + b^{\mathrm{gel}} \times \log_{10} d$ | | UVP-based particle flux counterparts $\log_{10} n_{\mathrm{uvp}}^{\mathrm{F}} = a^{\mathrm{uvp}} + b^{\mathrm{uvp}} \times \log_{10} d$ | |
|---|---|---|---|---|---|---|---|---|---|---|
| DST label | Station | Depth (m) | Index $k$ | POC flux (mg m$^{-2}$d$^{-1}$) | PON flux (mg m$^{-2}$d$^{-1}$) | Molar C:N | $a^{\mathrm{gel}}$ | $b^{\mathrm{gel}}$ | $a^{\mathrm{uvp}}$ | $b^{\mathrm{uvp}}$ |
| DST1 | D: M160-4-1 R: M160-12-1 | 100 | 1 | 78.69 | 9.20 | 9.97 | 9.85 | -2.11 | 10.49 | -2.20 |
| | | 200 | 2 | 47.76 | 5.81 | 9.58 | 9.45 | -2.03 | 10.29 | -2.18 |
| | | 400 | 3 | 48.97 | 5.19 | 10.99 | 9.81 | -2.14 | 10.80 | -2.38 |
| DST2 | D: M160-87-1 R: M160-87-1 | 100 | 4 | 65.48 | 13.20 | 5.78 | 9.29 | -1.90 | 10.87 | -2.37 |
| | | 200 | 5 | 35.68 | 6.68 | 6,22 | 9.80 | -2.15 | 10.74 | -2.35 |
| | | 400 | 6 | 38.86 | 5.34 | 8.48 | 9.51 | -2.00 | 10.48 | -2.20 |
| DST3 | D: M160-110-1 R: M160-116-1 | 100 | 7 | 105.43 | 12.67 | 9.70 | 10.02 | -2.21 | 12.44 | -2.83 |
| | | 200 | 8 | 49.41 | 5.31 | 10.84 | NA | NA | 10.67 | -2.27 |
| | | 400 | 9 | 39.25 | 4.11 | 11.13 | 9.54 | -2.03 | 10.78 | -2.36 |
| DST4 | D: M160-146-1 R: M160-164-1 | 100 | 10 | 68.03 | 14.92 | 5.31 | 9.43 | -1.95 | 10.50 | -2.22 |
| | | 200 | 11 | 58.60 | 5.94 | 11.49 | 10.23 | -2.29 | 9.78 | -1.95 |
| | | 400 | 12 | 60.29 | 7.94 | 8.84 | 10.51 | -2.47 | 11.26 | -2.56 |
| DST5 | D: M160-170-1 R: M160-182-1 | 100 | 13 | 52.03 | 7.39 | 8.20 | 9.96 | -2.16 | 10.37 | -2.18 |
| | | 200 | 14 | 42.81 | 5.54 | 9.00 | 11.13 | -2.69 | 10.83 | -2.41 |
| | | 400 | 15 | 59.11 | 6.29 | 10.95 | 10.47 | -2.44 | 10.50 | -2.28 |
| DST6 | D: M160-189-1 R: M160-198-1 | 100 | 16 | 76.17 | 9.51 | 9.33 | 8.61 | -1.65 | 9.60 | -1.88 |
| | | 200 | 17 | 46.73 | 5.57 | 9.78 | 9.11 | -1.85 | 10.0 | -2.07 |
| | | 400 | 18 | 44.00 | 5.28 | 9.70 | 9.51 | -2.04 | 9.66 | -1.95 |




**Table A3**. Particle size and sinking velocity data. Size (equivalent spherical diameter, ESD) and sinking velocity of in-situ formed aggregates collected during the M160 expedition. Measurements were conducted using the flow chamber system on board.

| Station | Size (mm) | Sinking Vel. (m d⁻¹) | Station | Size (mm) | Sinking Vel. (m d⁻¹) | Station | Size (mm) | Sinking Vel. (m d⁻¹) |
|---|---|---|---|---|---|---|---|---|
| M160-14-1 | 1.34 | 153.30 | M160-71-1 | 0.71 | 88.06 | M160-144-1 | 0.69 | 56.08 |
| M160-14-1 | 0.95 | 347.90 | M160-71-1 | 0.35 | 79.27 | M160-144-1 | 0.43 | 59.15 |
| M160-14-1 | 1.55 | 70.54 | M160-71-1 | 0.87 | 0.63 | M160-144-1 | 0.96 | 132.01 |
| M160-14-1 | 1.23 | 197.52 | M160-71-1 | 1.43 | 19.65 | M160-144-1 | 1.26 | 189.87 |
| M160-14-1 | 1.37 | 496.35 | M160-71-1 | 0.78 | 32.31 | M160-144-1 | 0.44 | 95.64 |
| M160-14-1 | 1.42 | 55.58 | M160-71-1 | 0.57 | 32.01 | M160-144-1 | 0.44 | 174.60 |
| M160-14-1 | 0.81 | 102.67 | M160-71-1 | 0.98 | 18.75 | M160-144-1 | 0.48 | 102.21 |
| M160-14-1 | 1.45 | 124.99 | M160-71-1 | 0.34 | 7.36 | M160-144-1 | 0.58 | 52.43 |
| M160-14-1 | 0.80 | 144.85 | M160-91-1 | 0.61 | 67.29 | M160-144-1 | 0.43 | 62.11 |
| M160-14-1 | 1.89 | 346.97 | M160-91-1 | 0.72 | 48.23 | M160-144-1 | 0.46 | 25.68 |
| M160-53-1 | 0.77 | 206.31 | M160-91-1 | 0.37 | 36.90 | M160-144-1 | 0.29 | 207.17 |
| M160-53-1 | 0.59 | 324.50 | M160-91-1 | 0.69 | 32.11 | M160-177-1 | 0.54 | 38.12 |
| M160-53-1 | 0.65 | 92.43 | M160-91-1 | 0.39 | 37.37 | M160-177-1 | 0.54 | 46.16 |
| M160-53-1 | 1.04 | 84.47 | M160-91-1 | 0.47 | 39.60 | M160-177-1 | 0.26 | 41.06 |
| M160-53-1 | 0.61 | 186.74 | M160-91-1 | 0.82 | 149.51 | M160-177-1 | 2.23 | 4.96 |
| M160-53-1 | 0.55 | 262.25 | M160-91-1 | 0.71 | 90.78 | M160-177-1 | 0.46 | 19.18 |
| M160-53-1 | 1.21 | 186.54 | M160-91-1 | 0.55 | 25.61 | M160-177-1 | 0.68 | 53.05 |
| M160-53-1 | 0.54 | 94.52 | M160-91-1 | 0.49 | 85.63 | M160-177-1 | 1.22 | 20.73 |
| M160-53-1 | 0.76 | 114.95 | M160-91-1 | 0.62 | 102.32 | M160-177-1 | 0.50 | 56.17 |
| M160-53-1 | 0.99 | 179.77 | M160-91-1 | 0.42 | 77.11 | M160-186-1 | 1.00 | 62.64 |
| M160-53-1 | 0.63 | 581.61 | M160-112-1 | 0.78 | 145.04 | M160-186-1 | 0.26 | 77.35 |
| M160-53-1 | 0.58 | 114.42 | M160-112-1 | 0.59 | 126.65 | M160-186-1 | 0.61 | 24.03 |
| M160-53-1 | 1.90 | 62.06 | M160-112-1 | 0.27 | 18.77 | M160-186-1 | 0.54 | 101.25 |
| M160-71-1 | 1.84 | 193.39 | M160-112-1 | 0.36 | 23.64 | M160-186-1 | 0.28 | 32.26 |
| M160-71-1 | 1.65 | 110.22 | M160-112-1 | 0.36 | 176.53 | M160-186-1 | 0.52 | 105.43 |
| M160-71-1 | 2.17 | 86.74 | M160-112-1 | 0.42 | 84.11 | M160-186-1 | 0.40 | 25.15 |
| M160-71-1 | 0.98 | 31.70 | M160-112-1 | 0.33 | 86.00 | M160-186-1 | 0.30 | 28.73 |
| M160-71-1 | 1.52 | 261.87 | M160-112-1 | 0.60 | 132.20 | M160-186-1 | 0.39 | 43.21 |
| M160-71-1 | 2.09 | 36.13 | M160-112-1 | 0.19 | 29.17 | | | |
| M160-71-1 | 0.39 | 11.72 | M160-112-1 | 0.36 | 41.97 | | | |



**Table A4**. Particle size and oxygen flux data. Size (equivalent spherical diameter, ESD) and interfacial diffusive flux of oxygen for in-situ formed aggregates collected during the M160 expedition. Measurements were conducted on board using the microsensor in the flow chamber.

| Station | Size (mm) | Diff. $O_2$ flux (nmol $O^2$ cm$^{-2}$h$^{-1}$) | Station | Size (mm) | Diff. $O_2$ flux (nmol $O^2$ cm$^{-2}$h$^{-1}$) | Station | Size (mm) | Diff. $O_2$ flux (nmol $O^2$ cm$^{-2}$h$^{-1}$) |
|---|---|---|---|---|---|---|---|---|
| M160-53-1 | 0.77 | 4.5327 | M160-53-1 | 0.55 | 4.8873 | M160-53-1 | 1.52 | 7.0449 |
| M160-53-1 | 1.04 | 3.1428 | M160-53-1 | 1.60 | 4.285 | M160-71-1 | 1.84 | 7.2257 |
| M160-53-1 | 0.61 | 2.3854 | M160-53-1 | 1.25 | 8.8444 | M160-71-1 | 1.65 | 6.9982 |
| M160-71-1 | 2.17 | 1.1999 | M160-71-1 | 0.98 | 4.8523 | M160-71-1 | 1.52 | 7.5312 |
| M160-71-1 | 2.09 | 0.8756 | M160-91-1 | 0.61 | 0.9358 | M160-91-1 | 0.72 | 1.4843 |
| M160-91-1 | 0.82 | 0.7346 | M160-91-1 | 0.71 | 0.2925 | M160-91-1 | 0.49 | 19.5579 |
| M160-91-1 | 0.62 | 1.5165 | M160-112-1 | 0.78 | 4.4208 | M160-112-1 | 0.59 | 3.5465 |
| M160-112-1 | 0.27 | 0.8509 | M160-144-1 | 0.69 | 8.0996 | M160-144-1 | 0.43 | 2.4215 |
| M160-144-1 | 0.96 | 5.1427 | M160-144-1 | 1.26 | 6.7506 | M160-144-1 | 0.48 | 8.9725 |
| M160-177-1 | 0.54 | 4.4655 | M160-177-1 | 0.54 | 6.834 | M160-186-1 | 1.0 | 3.0204 |
| M160-186-1 | 1.66 | 5.596 | | | | | | |




**Data availability**

All data used in this study are freely accessible. The UVP, CTD, ADCP, and drifter datasets can be respectively found at the following links:

https://doi.org/10.1594/PANGAEA.924375, https://doi.org/10.1594/PANGAEA.943432,

https://doi.org/10.1594/PANGAEA.943409, and https://doi.org/10.1594/PANGAEA.918612.

DST-based flux measurements, MSC-based size-sinking rates, and diffusive oxygen fluxes are available in the Appendix (Tables A2–4). The computer scripts used to produce the results are available from the corresponding authors upon request.

**Author contribution**

NN analysed the data, developed and implemented the presented methods and Gel-PISA package, prepared and visualized
the results, and wrote the manuscript. LH processed DST samples and measured biogeochemical fluxes. SR prepared the map of study region. CMF, LH, and NM performed in-situ sampling and measurements (MSC and DST) on board. RK contributed to processing of UVP profiles. TF contributed to the deployment of vmADCP on board, analysed the ADCP data and provided the position of the studied eddy. HH prepared the UVP5 for the cruise and instructed its use on board, downloaded and preprocessed the UVP5 data and contributed to postprocessing. AK conceptualized, planned and organized
the scientific programme of the multidisciplinary, multi-platform eddy studies during RV Meteor Cruise M160 and led the cruise. MHI planed the sampling strategy and supervised the study. All co-authors contributed to the discussion of results, critically reviewed and approved the manuscript.

**Competing interests**

The authors declare that they have no conflict of interest.

**Acknowledgements**

The authors extend gratitude to the captain and crew of RV Meteor M160, as well as to Juri Knudsen for operating the UVP5 on board.

**Financial support**

Funding from the German Federal Ministry of Education and Research (BMBF) through the following projects is greatly
acknowledged: REEBUS-WP6 (Grant 03F0815D), REEBUS-WP1 (Grant 03F0815A), Project 03F0629A, CUSCO-WP5 (Grant 03F0813A), and CO2MESO (Grant 03F0876A). Additional financial support was provided by the DFG-Research Center/Cluster of Excellence "The Ocean in the Earth System" (EXC-2077-390741603), and the Alfred Wegener Institute




Helmholtz Center for Polar Marine Research through the PoF IV program "Changing Earth - Sustaining our Future," Topic 6.3 of the German Helmholtz Association. NM and MHI acknowledge additional funding from the EU HORIZON-CL6
projects OceanICU (Grant 101083922) and SEA-Quester (Grant 101136480). CMF was funded by the AWI internal strategy fund EcoPump. RK acknowledges support from the French National Research Agency through the Make Our Planet Great Again program (ANR, #ANR-19-MPGA-0012) and funding from the German Science Foundation's Heisenberg Programme (#KI 1387/5-1).

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
