# Peer review of "A Novel Approach to Estimate Carbon and Nitrogen Flux from In Situ Optics: Application to Cyclonic Eddies off the Cape Verde Islands"

_EGUsphere, 2025_

## Author Comment (AC1)

Dear Reviewer,

Thank you very much for your constructive criticisms and thoughtful comments. In the following response, we clarify our original methodological approach and express our intention to implement the necessary revisions according to your recommendations.

**1. Clarification of the Study Focus**

We acknowledge that there is some ambiguity regarding whether our study should be viewed primarily as a methodological paper or as an investigation of the impact of eddy dynamics on carbon export. Our original intent was to introduce a novel formalism for estimating particulate organic carbon (POC) flux from vertical profiles of particle size distribution and abundance (PSDA) obtained using in situ camera systems (ICS). A key advantage of our approach (Method I) is that it can account for the contribution of suspended particles when estimating carbon flux from ICS-based PSDA profiles. Additionally, we incorporated bacterial degradation of organic carbon in settling particles as a function of water depth into the existing method for estimating POC flux attenuation from the ICS-based particle profiles (Method II). This enhancement accounts for particle age, temperature, and oxygen concentration as factors influencing carbon flux estimates. We also aimed to demonstrate the method's utility by applying it to cyclonic eddies and assessing their impact on POC export.

However, we recognize that we overextended the manuscript by including more observational data than is appropriate for a method paper and neglected sensitivity analyses. In the revision, we will refine the text, sharpen our focus on the methodological framework by incorporating a thorough sensitivity analysis, as you suggested, and relocating details on sampling techniques and observational results to the Appendix, or, if preferred, split the paper into two complementary parts.

**2. On the formulation of POC degradation and negative POC values**

Modeling microbial behavior in sinking marine particles is inherently challenging. This difficulty stems from the complex and poorly understood activity of aggregate-associated bacteria as they descend through highly variable in situ conditions, which can selectively influence degradation processes. Our primary formulation for the rate of change of POC in a settling aggregate (due to bacterial respiration) is a simplified model that assumes a constant degradation rate, $\lambda$ (mass of carbon respired per unit time), defined at reference temperature and oxygen concentration levels:

$$\frac{d\,\mathrm{POC}}{d\,t} = -\lambda\, f(T, O_2) \quad\Longrightarrow\quad \mathrm{POC} = \mathrm{POC}_0 - \lambda f(T, O_2)t,$$

where $POC_0$ is the particulate organic carbon mass at time $t = 0$, and $f(T, O_2)$ is a dimensionless modifier that accounts for the effect of temperature $T$ and oxygen concentration $O_2$ (relative to their reference values) on the degradation rate via a $Q_{10}$ formulation and Michaelis–Menten kinetics, respectively. Therefore, $\lambda f(T, O_2)$ represents the effective, depth-dependent rate at which POC is respired within a particle during settling. Assuming a power-law relationship between a particle's initial carbon mass and its size $d$, i.e., $\mathrm{POC}_0 = Ad^B$, the estimated POC flux at a given water depth reads:

$$\text{POC}^{\text{est.}} = \sum_i n_i^F [Ad_i^B - \lambda_i f(T, O_2) t_i],$$

where $n_i^F$ is the ISC-based particle flux in size class $i$, $A$ and $B$ are free parameters to be optimized, and $t_i$ is the particle travel time from the surface to the considered depth, estimated from the associated size–velocity relationship of settling particles.

We chose this formulation because $\lambda$ can be estimated directly from our bacterial $O_2$ respiration rate measurements on in situ collected aggregates via 1:1 conversion factor, where 1 mol of $O_2$ respired corresponds to 1 mol of $CO_2$ expired. We admit that, as particles age during settling (i.e., as $t$ increases), this simple linear model could—at great depths—predict unphysical negative POC values. **However, as explicitly noted in the manuscript (see lines 284–287 in the preprint), any negative POC mass is set to zero**, which is physically analogous to a scenario in which the particulate carbon of a marine particle is completely degraded at a specific depth. In practice, given the low values of $\lambda$ and the range of water depths and particle sinking velocities considered, the occurrence of negative values is unlikely. Note that low temperatures at great depths also significantly reduce the effective degradation rate $\lambda f(T, O_2)$ via $Q_{10}$ effect, further suppressing deep degradation. This also agrees with long term incubation experiments where it was shown that at 15°C the carbon-specific carbon respiration was ~13% per day and decreased to 2-3% per day at 4°C (Iversen & Ploug, 2013).

**3. Comparison with First-Order Kinetics**

We acknowledge that first-order kinetics—in which the rate of change of POC in a particle is assumed to be proportional to the amount of POC remaining— are theoretically more biologically grounded and commonly applied:

$$\frac{d\,\text{POC}}{d\,t} = -\lambda^* f(T, O_2)\text{POC} \quad \Rightarrow \quad \text{POC} = \text{POC}_0\, e^{-\lambda^* f(T,O_2)t}.$$

However, this model hinges on an accurate estimate of $\lambda^*$, the "carbon-specific" degradation rate (i.e., the fraction of POC degraded per unit time)- which our respiration data (expressed only as mass of carbon loss per time) do not permit.

Nonetheless, in response to the reviewer's comment and following DeVries & Weber (2017), we already implemented a first-order kinetics model to estimate the POC flux of settling particles at a given depth:

$$\text{POC}^{\text{est.}} = \sum_i n_i^F Ad_i^B e^{-\lambda^* f(T,O_2)t_i}$$

by treating $\lambda^*$ as a free parameter, optimized alongside $A$ and $B$ against our POC flux measurements. Preliminary calculations show that the optimized values of $A$ and $B$ differ only marginally from those obtained using the linear model, demonstrating that the choice between linear and first-order kinetic formulations does not significantly affect the final results. This consistency underscores the robustness of our methodology. We will discuss both the linear and first-order kinetic formulations in the revised manuscript. However, we are also prepared to reanalyze all results using the first-order model if required during the revision process.

**4. Language and Presentation**

We are also revising the manuscript's language to enhance clarity, narrative flow, and conciseness. We believe that some ambiguities may have arisen from the manuscript's length and organization, particularly the interchangeable mix of methodological descriptions and presentation of actual data, which constrained the inclusion of certain details. These issues will be addressed by restructuring the manuscript for better readability, with careful attention to details and formatting. By focusing primarily on the methodological framework, we aim to produce a much more concise and to-the-point manuscript.

We believe that these clarifications and our planned revisions will support a smooth review of the revised manuscript. Thank you again for your valuable feedback.

With best regards,
Corresponding Author

References:

Iversen, M. H. and Ploug, H.: Temperature effects on carbon-specific respiration rate and sinking velocity of diatom aggregates – potential implications for deep ocean export processes, Biogeosciences, 10, 4073–4085, 2013

DeVries, T. and Weber, T.: The export and fate of organic matter in the ocean: New constraints from combining satellite and oceanographic tracer observations, Global Biogeochem. Cycles, 31, 535–555, 2017

---

## Author Comment (AC2)

Dear Reviewer,

Thank you for your constructive comments and suggestions. Below, we outline our planned revisions in response to your recommendations.

In the revised manuscript, we will refine the text for greater clarity and conciseness, and sharpen our methodological framework by adding a comprehensive sensitivity analysis that traces how uncertainties (standard deviations) propagate through the analysis, as also requested by Reviewer 1. We will ensure consistent nomenclature and notation throughout and explicitly state all underlying assumptions to enhance transparency.

Regarding your remark—"*They are estimating the POC and PON flux based on particle size spectra without any recourse to measuring POC or PON. Anywhere that it is stated that these are observations, I think requires re-writing*"—we wish to clarify that, as explicitly mentioned in the manuscript (see lines 290–293 and 310–313 of the preprint), our model parameters used to estimate fluxes from in-situ camera system (ICS)–based particle size spectra (i.e., $A_{car.}$ and $B_{car.}$ for POC, and $A_{nit.}$ and $B_{nit.}$ for PON) were optimized against corresponding POC and PON fluxes measured using drifting sediment trap (DST) samples. In the manuscript, we used the term "observation" to refer specifically to DST-based flux measurements or to the particle size spectra acquired by the ICS, not to fluxes estimated from those spectra. However, we will carefully review the text and correct any instances where this wording might cause confusion.

We believe these revisions will strengthen the manuscript and welcome any further feedback you may have.

Best regards,

Corresponding author